# The SWI/SNF chromatin remodeling factor DPF3 regulates metastasis of ccRCC by modulating TGF-β signaling

Huanhuan Cui [1,2,3] ✉, Hongyang Yi[2,8], Hongyu Bao[2,4,8], Ying Tan[2,8], Chi Tian [2,8], Xinyao Shi[2], Diwen Gan [2], Bin Zhang [5], Weizheng Liang[2], Rui Chen [2], Qionghua Zhu[1,2], Liang Fang [1,2,3], Xin Gao [5], Hongda Huang [2,4], Ruijun Tian [6], Silke R. Sperling [7], Yuhui Hu [1,2] & Wei Chen [1,2,3] ✉

DPF3, a component of the SWI/SNF chromatin remodeling complex, has been associated with clear cell renal cell carcinoma (ccRCC) in a genome-wide association study. However, the functional role of DPF3 in ccRCC development and progression remains unknown. In this study, we demonstrate that DPF3a, the short isoform of DPF3, promotes kidney cancer cell migration both in vitro and in vivo, consistent with the clinical observation that DPF3a is significantly upregulated in ccRCC patients with metastases. Mechanistically, DPF3a specifically interacts with SNIP1, via which it forms a complex with SMAD4 and p300 histone acetyltransferase (HAT), the major transcriptional regulators of TGF-β signaling pathway. Moreover, the binding of DPF3a releases the repressive effect of SNIP1 on p300 HAT activity, leading to the increase in local histone acetylation and the activation of cell movement related genes. Overall, our findings reveal a metastasis-promoting function of DPF3, and further establish the link between SWI/SNF components and ccRCC.

Renal cell carcinoma (RCC) is the most common form of kidney cancer. It originates from renal tubular epithelial cells[1] and can be classified into three subtypes based on histological characteristics, of which clear cell renal cell carcinoma (ccRCC) is the most frequent one, making up about 80% of all RCC cases[2]. Over the past decades, significant progress has been made in the treatment of ccRCC with the development of targeted agents and immunotherapies[3]. However, most advanced ccRCC patients with metastases do not respond to these treatments and the long-term prognosis for these patients remains poor[4]. Therefore, further mechanistic dissection of ccRCC

development and metastasis will facilitate the discovery of new biomarkers as well as novel therapeutic strategies.

Genetic and epigenetic alterations play an important role in the development and progression of ccRCC. *VHL*, firstly described as a tumor suppressor in patients with the von Hippel-Lindau syndrome[5], is the most frequently mutated gene in ccRCC[6]. Loss of *VHL* results in increased expression of hypoxia-inducible transcription factors (HIFs) that drive dysregulated angiogenesis[7]. Next to *VHL*, *PBRM1*, a component of the SWI/SNF chromatin remodeling complex, is the second most commonly mutated gene in ccRCC[8]. It acts as a tumor suppressor

[1]Shenzhen Key Laboratory of Gene Regulation and Systems Biology, School of Life Sciences, Southern University of Science and Technology, Shenzhen 518005, China. [2]Department of Biology, School of Life Sciences, Southern University of Science and Technology, Shenzhen 518005, China. [3]Academy for Advanced Interdisciplinary Studies, Southern University of Science and Technology, Shenzhen 518005, China. [4]Key Laboratory of Molecular Design for Plant Cell Factory of Guangdong Higher Education Institutes, School of Life Sciences, Southern University of Science and Technology, Shenzhen 518005, China. [5]Computational Bioscience Research Center, Computer, Electrical and Mathematical Sciences and Engineering Division, King Abdullah University of Science and Technology (KAUST), Thuwal 23955-6900, Saudi Arabia. [6]Department of Chemistry, School of Science, Southern University of Science and Technology, Shenzhen 518005, China. [7]Cardiovascular Genetics, Charité-Universitätsmedizin Berlin, 13125 Berlin, Germany. [8]These authors contributed equally: Hongyang Yi, Hongyu Bao, Ying Tan, Chi Tian. ✉e-mail: cuihh@sustech.edu.cn; chenw@sustech.edu.cn

and regulates cell proliferation, migration and adhesion by modulating the expression of p21 and E-cadherin[8,9]. Moreover, *PBRM1* loss was associated with immune checkpoint blockade resistance by influencing interferon gamma-dependent signaling and tumor microenvironment[10]. In addition to *PBRM1*, other components of the SWI/SNF complex, including *SMARCA4* and *ARID1A*, are also found to be frequently altered in ccRCC. Mutated SWI/SNF components can function as either oncogenes or tumor suppressor genes[11], suggesting that mutations in this complex could result in altered epigenetic control of gene expression, thereby contributing to cancer development and progression[8,12]. Additional epigenetic regulators often mutated in ccRCC include the histone H3 lysine 36 trimethyltransferase *SETD2* and the deubiquitinating enzyme *BAP1*, which could function as tumor suppressors[13,14].

The human SWI/SNF chromatin remodeling complex consists of up to 15 components and is organized around a core catalytic subunit BRM or BRG1 (encoded by *SMARCA2* and *SMARCA4*, respectively)[11]. It utilizes the free energy of ATP hydrolysis to remodel chromatin from a condensed state to a transcriptionally accessible state, allowing transcription factors or other chromatin regulators to modulate gene expression. DPF3 belongs to the D4 protein family and is a noncatalytic subunit of the SWI/SNF complex. *DPF3* expresses two splice variants, *DPF3a* and *DPF3b*, which encode two distinct protein isoforms differing at their C-terminus (Fig. 1a). Due to its biased expression pattern across tissues, previous functional studies of DPF3 mainly focused on cardiac and skeletal muscle, neuron as well as brown adipose[15–19]. For

example, during the development of skeletal muscle, DPF3b recruits the SWI/SNF complex to target muscle-specific genes by recognizing methylated and acetylated histone tails via its PHD fingers[16,20], whereas DPF3a mediates its function in cardiac hypertrophy and skeletal muscle differentiation by interacting with transcription factors or other chromatin regulators through its specific C-terminus[17,19]. In contrast, only a couple of studies have linked DPF3 to human cancer. For instance, Hiramatsu et al. demonstrated that DPF3 could maintain stemness of glioma initiating cells and suggested it as a potential therapeutic target for glioblastoma[21]. In 2017, a genome-wide association study (GWAS) performed on 10,784 ccRCC patients and 20,406 controls identified one highly significant risk locus (rs4903064) in *DPF3* and, more importantly, found that the risk allele was associated with increased *DPF3* expression in two independent ccRCC cohorts[22,23]. More recently, Colli et al. and Protze et al. demonstrated that this SNP created a hypoxia-response element, which could upregulate *DPF3* expression and thereby affecting the proliferation rate of ccRCC cells in vitro[24,25]. However, so far, the functional role of DPF3 in other aspects of ccRCC progression has remained largely unexplored.

Here, we mechanistically dissect the function of DPF3 in ccRCC. First, by both positive and negative perturbation, we demonstrate that DPF3a but not DPF3b promotes kidney cancer cell migration both in vitro and in vivo. Further gene expression analysis reveals overexpression of DPF3a upregulates genes downstream of the TGF-β pathway, a signaling pathway with an established role in promoting cancer metastasis. The link to TGF-β pathways is through the SMAD

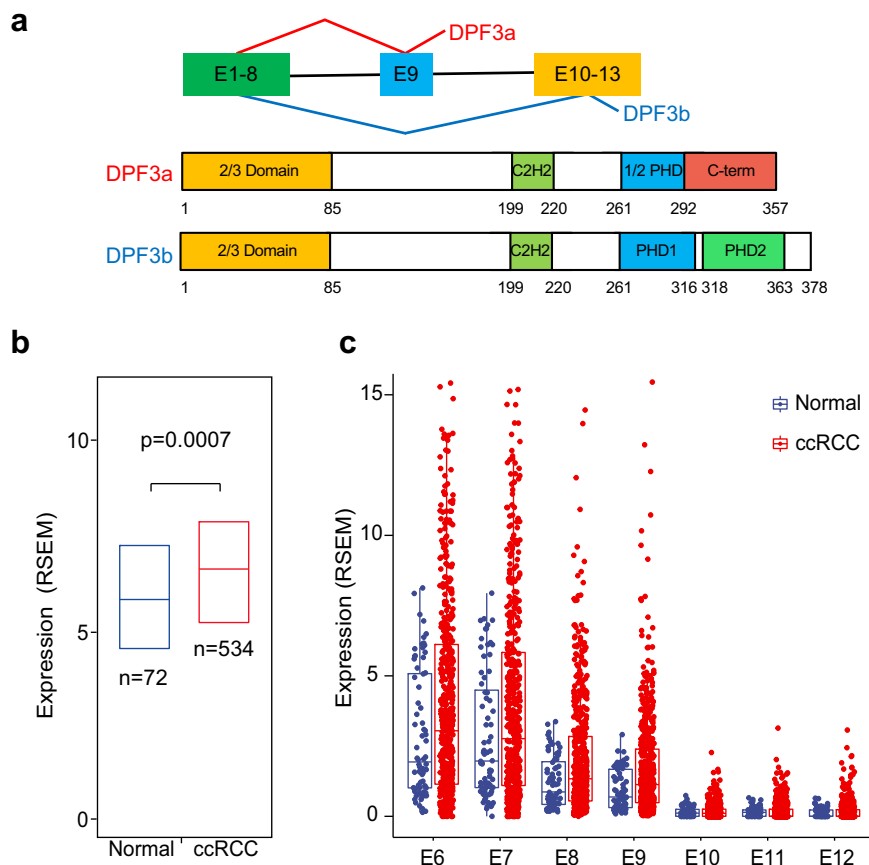

**Fig. 1 | Upregulated expression of *DPF3* in ccRCC. a** Schematic representation of the *DPF3* splicing isoforms and the corresponding protein structures. **b** Expression analysis of *DPF3* in normal kidney tissues (*n* = 72 individuals) and ccRCC patients from the TCGA-KIRC cohort (*n* = 534 individuals). Results represent median expression levels with 25 and 75% quartile. Y-axis represents the expression values of expectation-maximization (RSEM). Statistical significance was estimated using a two-sided Wilcoxon test. **c** Expression analysis of each exon of *DPF3* (exon 6 to exon 12) in normal kidney tissues (*n* = 72 individuals) and ccRCC patients of the TCGA-KIRC cohort (*n* = 534 individuals). Results represent median expression levels with 25 and 75% quartile. Each dot represents one individual. Y-axis represents the expression values of expectation-maximization (RSEM).

nuclear-interacting protein 1 (SNIP1), which is found to specifically interact with DPF3a and bridges DPF3a to SMAD4 and p300 histone acetyltransferase (HAT), the major transcriptional regulators in TGF-β pathway. Moreover, we demonstrate that the binding of DPF3a releases the repressive effect of SNIP1 on p300 HAT activity. This increases the local histone acetylation and subsequently upregulates the SMAD4 downstream target genes, including metastasis-promoting genes *SNAI1* and *MMP2*.

## Results

### The expression of *DPF3a* was upregulated in ccRCC

In a previous GWAS study of ccRCC, a risk locus has been identified in the first intron of *DPF3*[22]. Furthermore, it has been shown that the risk allele was associated with its increased expression. To examine the clinical relevance of *DPF3* expression in another independent dataset, we first resorted to the TCGA-KIRC cohort. As shown in Fig. 1b, its abundance was indeed significantly higher in the kidney tissues of ccRCC patients than in normal kidney tissues. We then separated the TCGA-KIRC patients into two groups based on the mutation status of *VHL* or the two SWI/SNF components *PBRM1* and *SMARCA4*. As shown in Supplementary Fig. 1a, the expression of *DPF3* was significantly higher only in the *VHL* mutant group, indicating that the upregulation of *DPF3* is likely associated with *VHL* mutations in ccRCC. Moreover, in another clinical cohort of ccRCC patients, the expression of *DPF3* was found to be significantly higher in metastatic tumors than in primary ones (Supplementary Fig. 1b), suggesting a potential role of *DPF3* in tumor metastasis. Given that *DPF3* expressed two distinct splicing isoforms (Fig. 1a), based on the same TCGA dataset, we then analyzed the expression of its exons in more detail, including exon 6–8 that are shared by *DPF3a* and *DPF3b*, exon 9 and exon 10–12 that were exclusively found in *DPF3a* and *DPF3b*, respectively. As shown in Fig. 1c, exon 9 had a comparable expression level as exon 6–8, whereas exon 10–12 were hardly detectable, indicating that *DPF3a* was the main isoform expressed and upregulated in ccRCC.

### DPF3a promoted kidney cancer cell migration in vitro and in vivo

Recently, Colli et al. reported that DPF3 overexpression could promote the proliferation of ccRCC cells[24] whereas Portze et al. showed that knockout in human urinary primary tubular cells had an opposite effect on cell proliferation in vitro[25]. To confirm whether DPF3 indeed played a role in ccRCC cell proliferation, we knocked down DPF3 using RNA interference in a primary ccRCC cell line[26], in which DPF3 was expressed (Supplementary Fig. 2a), and measured cell proliferation using CCK-8 assays. Compared to a non-targeting control siRNA (siNon), successful reduction of DPF3 expression (Supplementary Fig. 2b) could significantly inhibit cell proliferation (Supplementary Fig. 2b) at 72 h after seeding. To explore the distinct function of the two isoforms in regulating cancer cell proliferation, we overexpressed the two isoforms separately in 786-O (Supplementary Fig. 2d, e), one of the first established ccRCC cell lines, in which DPF3 is almost undetectable (Supplementary Fig. 2a). We observed that the overexpression of DPF3a (DPF3a-OE), but not DPF3b could significantly enhance 786-O cell proliferation at 72 h after seeding (Supplementary Fig. 2f). Moreover, to clarify the impacts of DPF3a-OE on tumor growth in vivo, a subcutaneous xenotransplanted tumor model was established with 786-O cells. Compared with the control group, DPF3a-OE cells exhibited significant tumor growth-promoting effects in vivo (Supplementary Fig. 2g, h).

To further examine whether DPF3 played any role in ccRCC cell migration and metastasis, we measured cell migration/invasion by carrying out Transwell assays after DPF3 perturbation. We found that cell migration/invasion abilities were significantly reduced in the primary ccRCC cells upon DPF3 knockdown (Fig. 2a), whereas only DPF3a-OE in 786-O cells could increase migration/invasion capabilities at 12 to 24 h, respectively, after seeding (Fig. 2b). To investigate whether DPF3a affected cell migration in vivo, we took advantage of a mouse metastasis model, in which tumor cells were injected into the spleens, and metastasis was then measured in the livers of immunodeficient mice[27]. Consistent with its ability on promoting migration in vitro, DPF3a-OE notably enhanced 786-O cell metastasis in vivo (Fig. 2c–e). Taken together, these results demonstrated that DPF3a promoted cell migration and tumor metastasis both in vitro and in vivo. Given the much stronger effect of DPF3a overexpression in cell migration, we hereafter focused on its effect on cell migration.

### DPF3a regulated the expression of genes associated with cellular movement

To understand molecular mechanisms underlying the increased migration potential upon DPF3a-OE, we compared the gene expression profile of DPF3a-OE to the control cells expressing empty vector (EV) using RNA-seq. Two biological replicates were carried out for each cell population. As shown in Fig. 3a, upon DPF3a-OE, a total of 1,713 genes were differentially expressed (FDR <0.05, |Log2 (Fold change)| >0.58), with 877 genes upregulated and 836 genes downregulated in DPF3a-OE cells, respectively (Supplementary Data 1). Interestingly, these differentially expressed genes were highly enriched with functions related to "cellular movement" ($p$ value <$10^{-34}$) (Fig. 3b). For instance, key regulators of epithelial-mesenchymal transition (EMT), including *SNAI1*, *ZEB1*, *MMP1*, *MMP2*, *COL1A1*, and *WNT7B*, were upregulated in DPF3a-OE cells (Supplementary Fig. 3a), in accordance with their high metastatic potential. As shown in Fig. 3c, the upregulation of the two key EMT transcription factors SNAI1 and ZEB1 could also be observed at the protein level (Fig. 3c). It has been shown that MMP2 could promote cell migration/invasion in an autocrine manner by degrading extracellular matrix. Therefore, in addition to validate the increased level of MMP2 protein (Fig. 3c), we also measured the activity of secreted MMP2 by carrying out a fluorometric enzymatic activity assay. As shown in Fig. 3d, the activity of secreted MMP2 was indeed increased in DPF3a-OE cells (Fig. 3d).

Using Ingenuity Pathway Analysis (IPA) tool, we then predicted the upstream regulator of these differentially expressed genes. It turned out that TGFB1, a well-known EMT modulator[28], was one of the top candidates (Fig. 3e), suggesting that DPF3a might be involved in regulating TGF-β signaling pathways. In consistent with its well-known EMT promoting function, the treatment of 786-O cells with recombinant TGFB1 could promote cell migration/invasion (Fig. 3f), and increase the expression of cell movement-related genes, which were also upregulated by DPF3a-OE (Supplementary Fig. 3b). To check whether DPF3a could influence the expression of TGF-β ligands, we quantified autocrine production of TGFB1 in the culture medium of 786-O cells with and without DPF3a overexpression using an ELISA assay. As shown in Supplementary Fig. 3c, the amount of TGFB1 was not altered upon DPF3a-OE. These data together suggested that (1) DPF3a regulated cell movement via TGF-β signaling pathway and (2) but the regulation was not at the level of TGF-β ligands.

Given that DPF3 is a noncatalytic component of the SWI/SNF chromatin remodeling complex, we also investigated the effect of DPF3a overexpression on chromatin accessibility by performing the assay for transposase-accessible chromatin with high-throughput sequencing (ATAC-seq) in DPF3a-OE and control 786-O cells. Surprisingly, the global chromatin accessibility showed high similarity between DPF3a-OE and control 786-O cells (Pearson correlation: 0.98, Supplementary Fig. 3d). Even ATAC-seq signals at promoters of genes differentially expressed upon DPF3a manifested no significant difference between DPF3a-OE and control cells (Pearson correlation: 0.99, Fig. 3g), indicating that DPF3a regulated its downstream targets without altering chromatin accessibility.

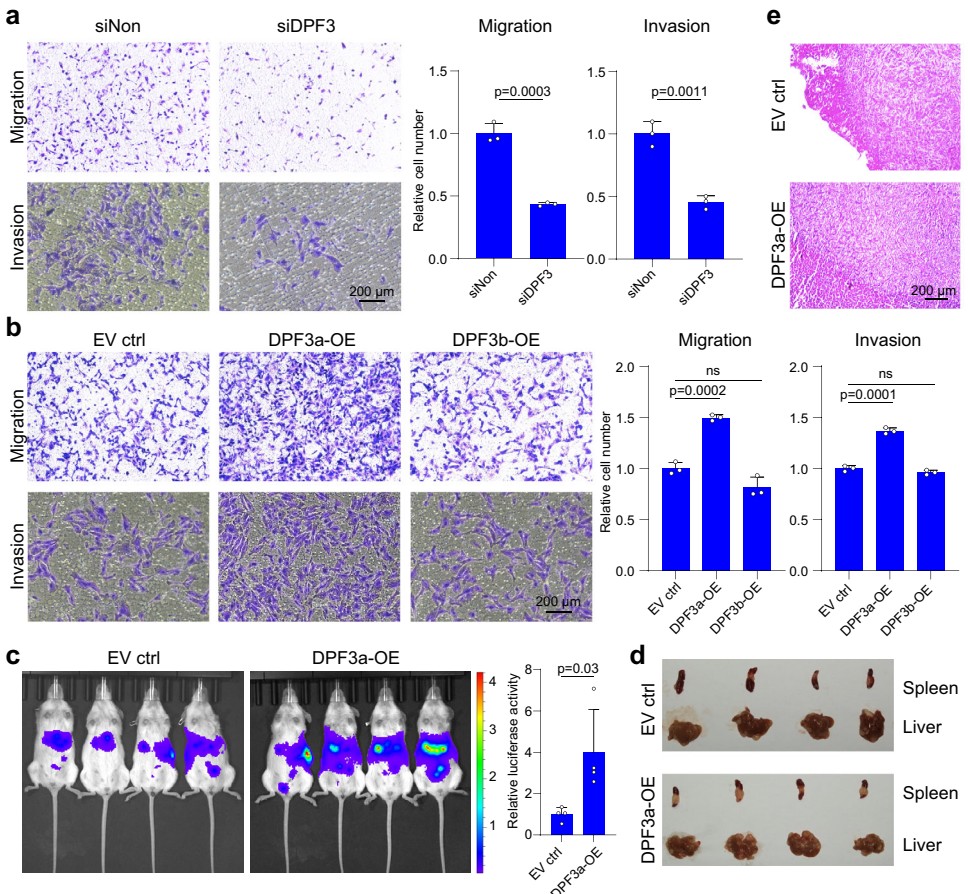

**Fig. 2 | DPF3a promoted cell migration in vitro and in vivo. a** The migration and invasion ability of primary ccRCC cells with or without DPF3 knockdown was measured using the Transwell assay. Cells were seeded on top of the membrane and fixed and stained with crystal violet after 24 h incubation. Cell numbers on the bottom of the membrane were quantified and statistical significance was estimated using a two-sided student *t*-test. Data were presented as mean values ± SD (*n* = 3 independent experiments). Scale bar, 200 μm. **b** The migration and invasion ability of 786-O cells with or without DPF3a/b overexpression was measured using Transwell assay. Cells were seeded on top of the membrane and fixed and stained with crystal violet after 24 h incubation. Cell numbers on the bottom of the membrane were quantified and statistical significance was estimated using a two-sided student *t*-test. Data were presented as mean values ± SD (*n* = 3 independent experiments). Scale bar, 200 μm. **c** The meta-static ability of 786-O cells with and without DPF3a overexpression was mea-sured using a mouse metastasis model. Tumor cells expressing luciferase were visualized with D-Luciferin as substrate using an in vivo imaging system. Total luminescence intensity was quantified and statistical significance was estimated using a two-sided student *t*-test. Data were presented as mean values ± SD (*n* = 4 animals). **d** The images of the liver and spleen were dissected from mice with metastasis. **e** Representative images of Hematoxylin and Eosin (H&E) staining of metastatic lesions in livers (*n* = 4 animals). Scale bar, 200 μm. Source data are provided in the Source Data file.

### DPF3a interacted with SNIP1, a SMAD nuclear-interacting protein

Previously, we and others have demonstrated that DPF3a achieved its transcriptional regulation via interaction with cofactors[17,19]. Therefore, to understand how DPF3a could activate TGFB1 downstream genes, we searched for DPF3a interaction partners by carrying out immunopre-cipitation followed by mass spectrometry (IP-MS) analysis. As shown in Table 1, we identified a total of 27 proteins as DPF3a interaction part-ners in 786-O cells, including 11 out of 15 known subunits of the SWI/SNF chromatin remodeling complex, which demonstrated the high efficiency and specificity of our IP-MS experiment. Beyond the com-ponents of the SWI/SNF complex, the SMAD nuclear-interacting pro-tein 1 (SNIP1) drew our attention as it has been reported as a repressor of SMAD-dependent TGF-β signaling[29].

We then validated the interaction between DPF3a and SNIP1 by co-immunoprecipitation (Co-IP) in DPF3a-OE cells using antibodies against HA-tagged DPF3a and endogenous SNIP1, respectively. As shown in Fig. 4a, in both forward and reverse Co-IP followed by immunoblotting with HA antibody and SNIP1 antibody, we could demonstrate that DPF3a specifically interacted with SNIP1 (Fig. 4a). In vitro GST pulldown assays using recombinant proteins also confirmed that DPF3a and SNIP1 physically interacted with each other (Supple-mentary Fig. 4a). Furthermore, we checked the localization of the two proteins using immunofluorescence staining and observed a sig-nificant co-localization between the two proteins in the nucleus (Fig. 4b). To check whether SNIP1 binds directly to DPF3a or via other components of the SWI/SNF complex, we carried out SNIP1 IP-MS in 786-O cells with and without DPF3a overexpression. As shown in Supplementary Table 1, overexpression of DPF3a strikingly enhanced the binding affinity of SNIP1 to the other components of the SWI/SNF complex, suggesting that SNIP1 was more likely directly associated with DPF3a than through the other components of the SWI/SNF complex.

SNIP1 has a predicted intrinsic disorder region (IDR) at N-terminus and a Forkhead-associated (FHA) domain at C-terminus (Supplemen-tary Fig. 4b). To map the region of SNIP1 interacting with DPF3a, we co-transfected Flag-tagged full-length SNIP1 (FL), the IDR (amino acid, a.a. 1–240) and the C-terminus (a.a. 236–396) (Supplementary Fig. 4c) together with HA-tagged DPF3a in HEK293T cells, respectively. Using Co-IP analysis, we observed that DPF3a interacted with SNIP1-FL and SNIP1-IDR but not the C-terminus of SNIP1 (Fig. 4c). In vitro GST pull-down assay also confirmed the direct interaction between SNIP1-IDR

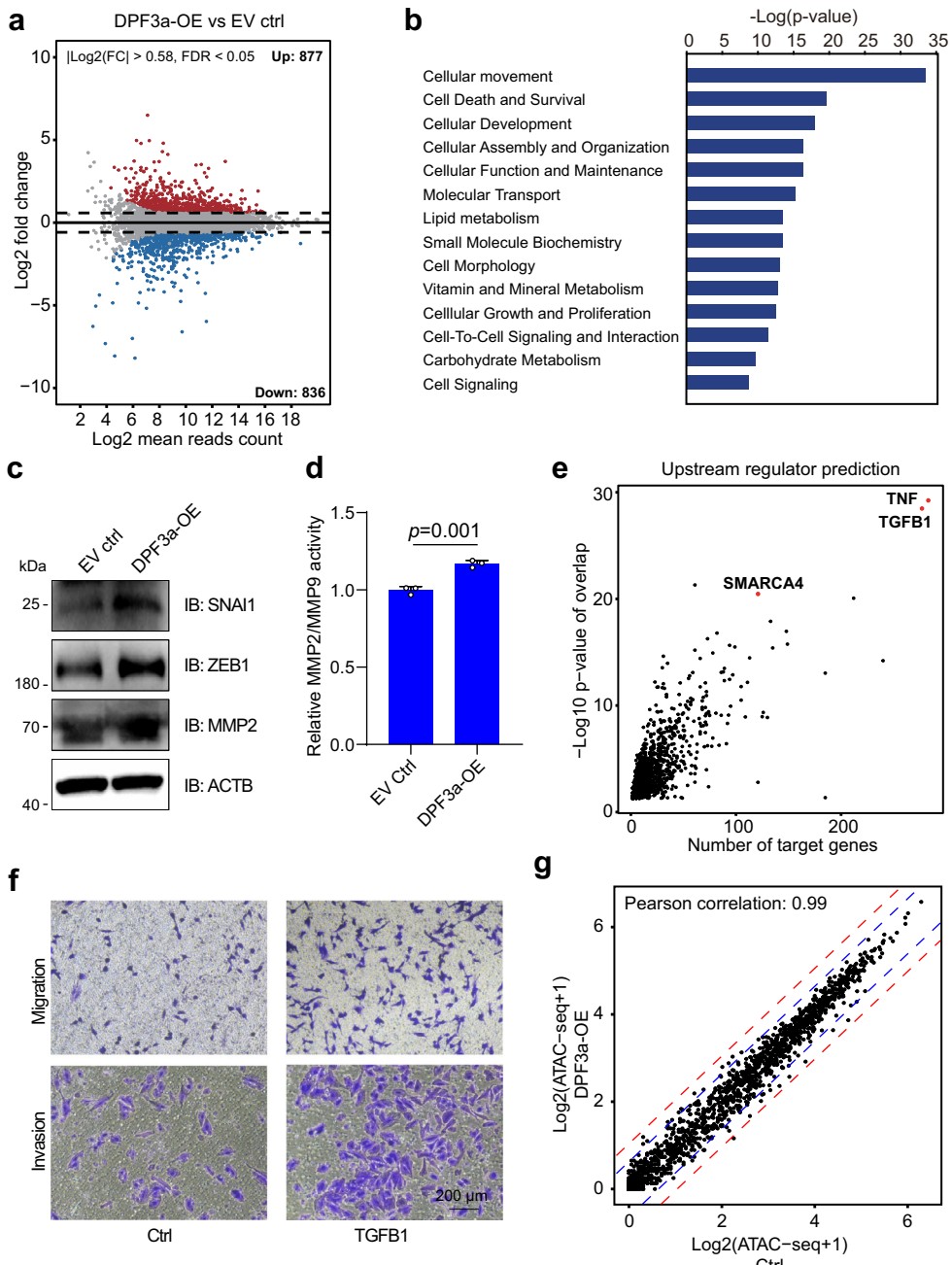

**Fig. 3 | Comparison of transcriptome and chromatin accessibility between 786-O cells with and those without DPF3 expression. a** The MA plot comparing gene expression profile of 786-O cells with and those without DPF3a expression. The x-axis and y-axis represent Log2 (mean read counts) and Log2 (fold change), respectively. Red dots indicate upregulated genes, whereas blue dots indicate downregulated genes. The results represent two independent biological replicates. **b** The "Diseases & Functions" analysis of genes upregulated after DPF3a over-expression using Ingenuity Pathway Analysis (IPA) software (QIAGEN). The *p* value was calculated using the right-tailed Fisher's exact test. The results represent two independent biological replicates. **c** The protein level of EMT-related genes SNAI1, ZEB1, and MMP2 in DPF3-OE and control 786-O cells was analyzed by immuno-blotting (IB). Three independent experiments were performed and similar results were obtained. **d** Effect of DPF3a overexpression on MMP2 activity in 786-O cells. MMP2 activity was determined using a fluorogenic substrate as described in Methods. Data were presented as mean values ± SD (*n* = 3 independent

experiments). Statistical significance was estimated using a two-sided student *t*-test. **e** The "Upstream Regulator" analysis of genes upregulated after DPF3a overexpression using IPA. The *p* value was calculated using the right-tailed Fisher's exact test. The x-axis represents the number of DPF3a regulated genes which were predicted to be downstream targets of specific regulators. The y-axis represents −Log10 *p* value of overlap between DPF3a regulated genes and downstream targets genes of a specific regulator. The results represent two independent biological replicates. **f** The migration and invasion ability of 786-O cells after TGFB1 treatment (20 ng/ml) was measured using Transwell assay. Three independent experiments were performed and similar results were obtained. **g** Scatter plot comparing ATAC-seq normalized read counts (CPM counts per million) at the promoters of genes differentially expressed in 786-O cells versus those without DPF3a overexpression. Pearson's correlation test was performed. The blue and red dash line represents the fold change of 1.5 and 2, respectively. The results represent one biological replicate. Source data are provided in the Source Data file.

and DPF3a (Supplementary Fig. 4a). Next, we went on to map the region of DPF3a interacting with SNIP1. As there was no interaction observed between DPF3b and SNIP1 in our Co-IP assay (Fig. 4a), as well as DPF3a and DPF3b were only different in the C-terminus, we assumed that the DPF3a-SNIP1 interaction was mediated by the DPF3a-specific C-terminus. To confirm this, we performed Co-IP assays in HEK293T cells co-overexpressing HA-tagged FL, N-terminus (a.a. 1–250), and C-terminus (a.a. 254–357) of DPF3a (Supplementary Fig. 4d) with Flag-tagged SNIP1-FL and found that SNIP1-FL indeed interacted with the C-terminal but not N-terminal region of DPF3a (Fig. 4d). Interestingly, in the in vitro pulldown assay using recombinant proteins, we observed that both DPF3a N-terminus and C-terminus could directly interact with SNIP1-IDR (Supplementary Fig. 4e). Collectively, all these data demonstrated that the two proteins physically interacted via C-terminus of DPF3a and SNIP1-IDR. The N-terminus of DPF3a, known to be bound by BRG1/BRM[30,31], was likely inaccessible for SNIP1 interaction in vivo.

## DPF3a and SNIP1 co-regulated cell migration via transcriptional regulation

Previous work has shown that SNIP1 functions as a repressor of TGF-β signaling by interfering with the interaction of p300 with the activated SMAD complex[29]. However, the effect of SNIP1 on ccRCC metastasis has not been reported yet. To clarify this, we performed both knockdown and overexpression of SNIP1 in 786-O cells (Supplementary Fig. 5a, b). As shown in Fig. 5a, b and Supplementary Fig. 5c, inhibition of SNIP1 could significantly enhance cell migration, whereas over-expression of SNIP1 had the opposite effect. Notably, the perturbation of SNIP1 caused no effect on cell proliferation (Supplementary Fig. 5d). Consistent with in vitro data, we observed a significant increase and slight decrease (statistically non-significant) of metastatic capacity upon SNIP1 knockdown and overexpression, respectively, in the in vivo model (Fig. 5c, d). Then, to examine whether DPF3a exerted its effect via SNIP1, we overexpressed DPF3a in SNIP1 knockdown and over-expression cells (Supplementary Fig. 5b), respectively and then checked cell migration abilities. We found that DPF3a overexpression could partially rescue migration ability inhibited by SNIP1 over-expression (Fig. 5a). Importantly, we did not observe a further enhancement of migration when we overexpressed DPF3a in SNIP1 knockdown cells (Fig. 5b), suggesting that DPF3a promoted cell migration via relieving the inhibitory effect of SNIP1.

To further mechanistically understand the regulation mediated by the interaction between SNIP1 and DPF3a, we carried out RNA-seq in 786-O cells with and without SNIP1 knockdown. In total, we identified a total of 824 differentially expressed genes upon SNIP1 knockdown (FDR <0.05, |Log2 (Fold change)| >0.58) (Fig. 5e), with 454 genes upregulated and 370 genes downregulated, respectively (Supplementary Data 2). We then correlated the differentially expressed genes in SNIP1 knockdown with those in DPF3a overexpression experiments and found a list of 286 genes commonly regulated by the perturbation of SNIP1 and DPF3a. As expected from the negative impact of DPF3a on SNIP1-mediated regulation, 74% (212 out of 286) of these genes were regulated in a consistent manner between the two conditions (Supplementary Fig. 5e). Again, these genes were significantly enriched for GO terms related to "cellular movement" (Supplementary Fig. 5f), suggesting that DPF3a and SNIP1 co-regulated a subset of cell migration-related genes.

Both DPF3a and SNIP1 are known chromatin-associated proteins. To examine the co-occupancy of DPF3a and SNIP1 across the genome, we performed chromatin immunoprecipitation followed by sequencing (ChIP-seq) of HA-tagged DPF3a and endogenous SNIP1 in 786-O cells overexpressing HA-tagged DPF3a. In total, we obtained 51,530 binding sites for DPF3a, of which 12.89%, 48.87%, and 38.22% were at the promoter, gene-body, and intergenic region,

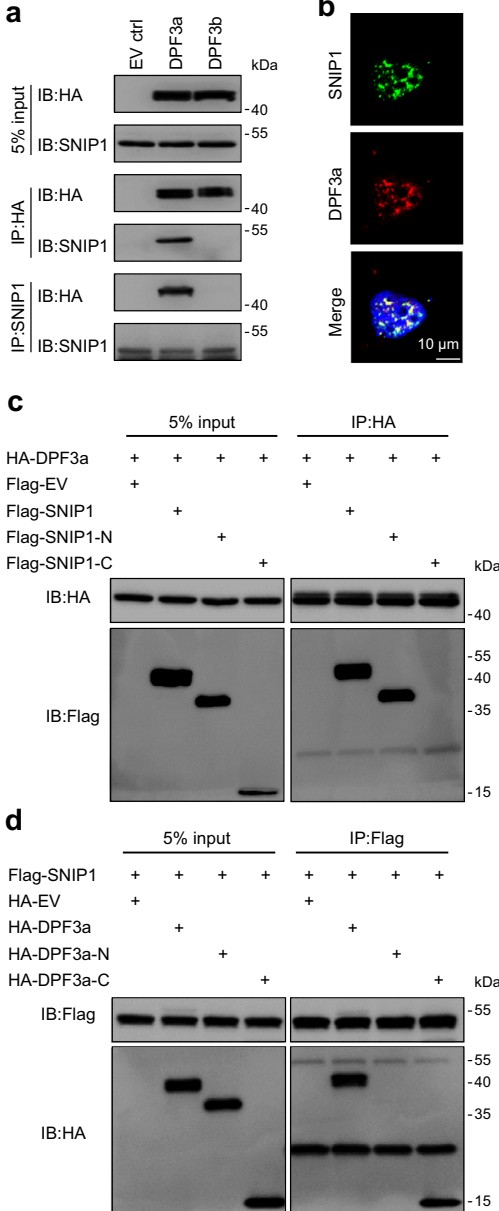

**Fig. 4 | Interaction between DPF3a and SNIP1. a** DPF3a interacted with SNIP1. Immunoprecipitation was carried out using antibody against HA or SNIP1, followed by immunoblotting (IB) to detect endogenous SNIP1 using anti-SNIP1 antibody or HA-tagged DPF3a/b using the anti-HA antibody in 786-O cells expressing empty vector, HA-tagged DPF3a or HA-tagged DPF3b. Input lanes represents 5% of total protein lysate. **b** Co-localization of DPF3a and SNIP1 in the nucleus. 786-O cells expressing HA-tagged DPF3a were fixed with methanol and stained with antibodies against HA (red) and SNIP1 (green). Scale bar, 10 μm. **c** DPF3a interacted with the N-terminal domain of SNIP1. HA-tagged DPF3a were co-transfected separately with Flag-tagged full-length SNIP1 or its C-/N- terminus in HEK293T cells. HA-tagged DPF3a was purified by immuno-precipitation with anti-HA magnetic beads, followed by immunoblotting to detect Flag-tagged SNIP1 truncations with the anti-Flag antibody. **d** SNIP1 interacted with the C-terminus of DPF3a. Flag-tagged SNIP1 were co-transfected separately with full-length HA-tagged DPF3a or its C-/N- terminus in HEK293T cells. Flag-tagged SNIP1 was purified by immunoprecipitation with anti-Flag magnetic beads, followed by immunoblotting to detect HA-tagged DPF3a with the anti-HA antibody. **a–d** Three independent experiments were performed and similar results were obtained. Source data are provided in the Source Data file.

**Table 1 | DPF3a interaction partners were identified using immunoprecipitation followed by mass spectrometry (IP-MS)**

| Gene names | Peptides | Coverage | LogFC LFQ | Protein names |
|---|---|---|---|---|
| SMARCE1 (BAF57) | 15 | 43.6 | 8.47 | SWI/SNF complex subunit SMARCE1 |
| SMARCD2 (BAF60B) | 14 | 39.5 | 8.31 | SWI/SNF complex subunit SMARCD2 |
| ARID1A (BAF250A) | 26 | 16.4 | 8.22 | AT-rich interactive domain-containing protein 1A |
| SMARCA4 (BRG1) | 20 | 14 | 8.19 | SWI/SNF complex subunit SMARCA4 |
| SMARCB1 (BAF47) | 9 | 25.5 | 8.19 | SWI/SNF complex subunit SMARCB1 |
| ACTL6A (BAF53A) | 7 | 21.4 | 7.82 | Actin-like protein 6A |
| SMARCD1 (BAF60A) | 11 | 27.8 | 7.78 | SWI/SNF complex subunit SMARCD1 |
| SMARCA2 (BRM) | 15 | 11.3 | 7.35 | SWI/SNF complex subunit SMARCA2 |
| SMARCD3 (BAF60C) | 11 | 27.7 | 7.35 | SWI/SNF complex subunit SMARCD3 |
| LCN1P1; LCN1 | 2 | 11.1 | 7.28 | Putative lipocalin 1-like protein 1;Lipocalin 1 |
| LMNA | 28 | 60.4 | 7.20 | Lamin A/C |
| TFB1M | 4 | 13.9 | 7.08 | Dimethyladenosine transferase 1, mitochondrial |
| ASF1B | 2 | 19.6 | 7.08 | Histone chaperone ASF1B |
| TUBA4A | 11 | 34.4 | 7.06 | Tubulin alpha-4A chain |
| DBT | 2 | 6.2 | 7.04 | Dihydrolipoamide Branched Chain Transacylase |
| LYZ | 2 | 20.2 | 7.03 | Lysozyme; Lysozyme C |
| U2SURP | 2 | 7.9 | 6.96 | U2 snRNP-associated SURP motif-containing protein |
| CSNK1A1 | 3 | 8.2 | 6.94 | Casein kinase I isoform alpha |
| YWHAQ | 4 | 18.4 | 6.85 | 14-3-3 protein theta |
| MRPL37 | 3 | 8.7 | 6.79 | 39 S ribosomal protein L37, mitochondrial |
| PYGB;PYGM | 3 | 5.1 | 6.70 | Glycogen phosphorylase, brain form/muscle form |
| PFKL | 3 | 5.8 | 6.62 | ATP-dependent 6-phosphofructokinase, liver type |
| CTPS1 | 2 | 6.4 | 6.58 | CTP synthase 1 |
| SNIP1 | 2 | 7.1 | 6.34 | Smad nuclear-interacting protein 1 |
| DNAJB4 | 2 | 7.1 | 5.82 | DnaJ homolog subfamily B member 4 |
| SMARCC2 (BAF170) | 36 | 29.2 | 2.73 | SWI/SNF complex subunit SMARCC2 |
| SMARCC1 (BAF155) | 36 | 36.9 | 2.19 | SWI/SNF complex subunit SMARCC1 |

DPF3a interaction partners were identified using immunoprecipitation followed by mass spectrometry (IP-MS). A total of 27 proteins were identified with a minimum number of two peptides and 5% coverage, as well as the Log10 fold change of a label-free quantity (LFQ) >2.

respectively (Supplementary Fig. 5g). For SNIP1, 22,791 binding sites were identified, with 30.13, 40.44, and 29.39% located at the promoter, gene-body, and intergenic region, respectively (Supplementary Fig. 5h). As expected, we observed a significant overlap of binding sites (16,575 co-binding sites) between DPF3a and SNIP1 (Supplementary Fig. 5i), suggesting genome-wide co-occupancy of DPF3a and SNIP1. Among the 16,575 common peaks, 28.82% and 30.35% occurred at gene promoters and intergenic regions, respectively (Supplementary Fig. 5i). As shown in Fig. 5f, the peak intensity of DPF3a and SNIP1 co-binding sites was significantly higher than those exclusively bound with either DPF3a or SNIP1, suggesting their stronger chromatin association at these co-binding sites. In line with this, we also observed higher accessible chromatin at DPF3a-SNIP1 co-binding regions based on our ATAC-seq data (Supplementary Fig. 5j). As four representative examples shown in Fig. 5g, both DPF3a and SNIP1 bound at the promoter of *SNAI1* and *COL1A1*, and at the intronic regions of *MMP2* and *WNT7B*. In line with the binding pattern, the expression of the four EMT-related genes were upregulated upon SNIP1 knockdown or DPF3a overexpression (Fig. 5g), indicating that DPF3a and SNIP1 regulated the expression of their targets through binding at either proximal or distal cis-regulatory sites.

To further clarify which of the two bound to DNA and then recruited the other, we performed ChIP-qPCR to check DPF3a binding at DPF3a-SNIP1 common targets in DPF3a-OE cells with or without SNIP1 knockdown. As shown in Fig. 5h, a significant reduction of DPF3a enrichment was observed upon SNIP1 knockdown (Fig. 5h). In contrast, the SNIP1 enrichment at these targets was not significantly altered between the 786-O cells with and

without DPF3a overexpression (Fig. 5i). These results demonstrated that it was SNIP1 that recruited DPF3a to their common target sites.

SNIP1 has been recently identified as a core member of the retention and splicing complex (RES), which also consists of the U2 snRNP-associated protein RBMX2 and the bud site-selection protein BUD13[32]. It has been shown that genetic loss of RES components in zebrafish caused neurodevelopmental disorders and cell death. The defects were mainly due to widespread intron mis-splicing, with 74–79% and 7–9% of introns showing increased and decreased retention in the RES mutants, respectively[33]. Therefore, to check whether our SNIP1 perturbation in 786-O cells could result in a similar splicing defect, we compared SNIP1 knockdown and control samples with the five different types of alternative splicing events, including skipped exon (SE), alternative 5′ splice site (A5SS), alternative 3′ splice site (A3SS), mutually exclusive exons (MXE) and retained intron (RI). In total, 724 differential splicing events were identified (Supplementary Fig. 6a). In contrast to previous observations in zebrafish early embryos, we did not observe a substantial increase in intron retention after SNIP1 knockdown in 786-O cells. In addition, we also analyzed the differential splicing events caused by DPF3a overexpression. A total of 2138 events were identified, including 1476 SE, 155 A5SS, 203 A3SS, 108 MXE, and 196 RI (Supplementary Fig. 6b). Of these, only 29 events from 26 genes were also differentially spliced in a consistent manner upon SNIP1 knockdown (Supplementary Fig. 6c and Supplementary Data. 3). Importantly, splicing of these genes has not been reported to affect cell migration. Therefore, the observed effect of SNIP1 and DPF3a on cell migration was mainly through transcriptional regulation instead of splicing.

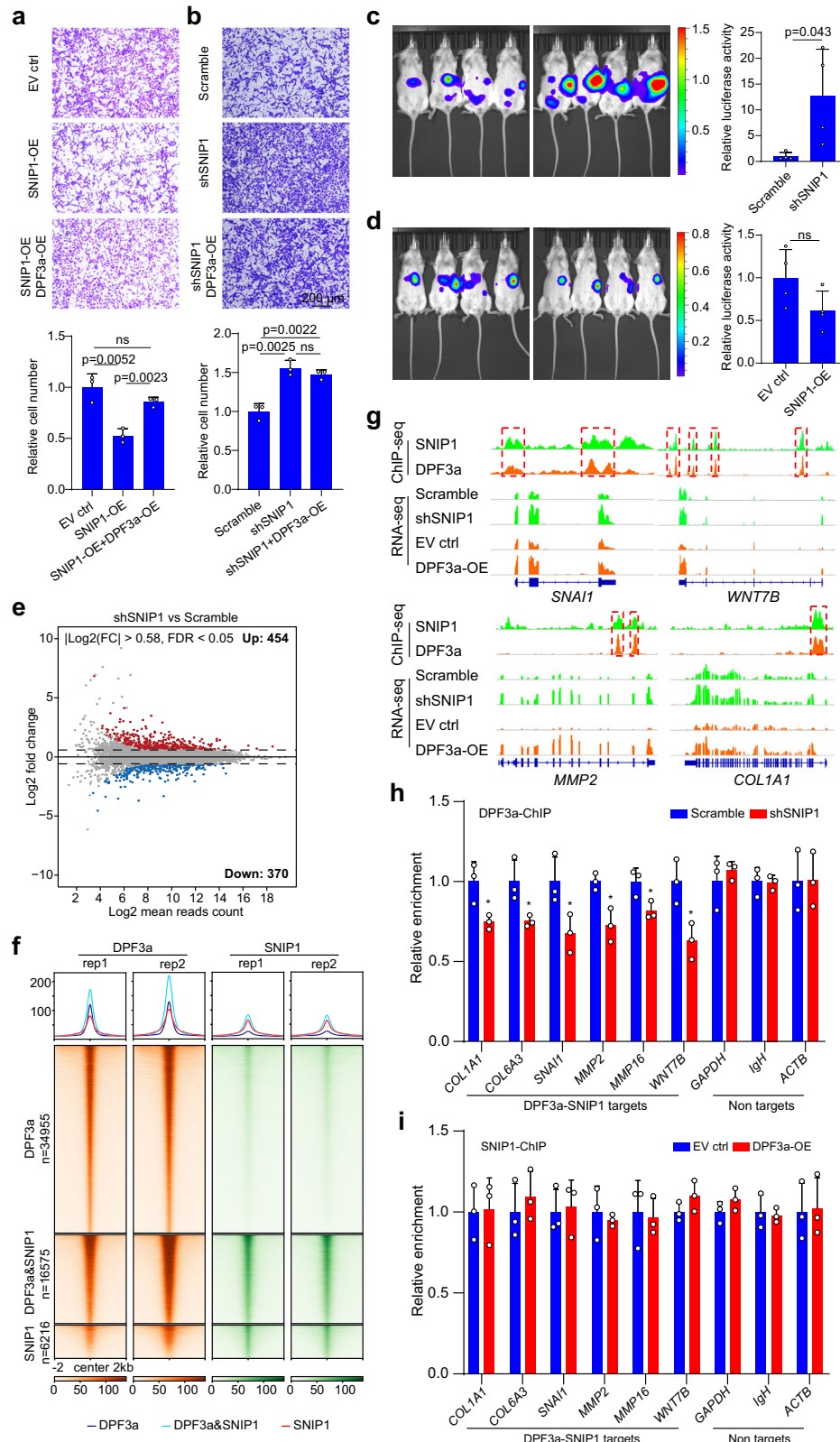

## DPF3a binding released the inhibitory effect of SNIP1 on p300 activity

SNIP1 has been reported to interact with both SMAD4 and the histone acetyltransferase p300, and the binding of SNIP1 could then inhibit the SMAD4-associated p300 activity[29,34]. Given that DPF3a could bind to SNIP1 and activate TGF-β downstream genes, we examined the interaction between DPF3a and SNIP1, SMAD4 as well as p300. For this

purpose, we performed Co-IP assays in 786-O cells expressing HA-tagged DPF3a using antibodies against HA. As shown in Fig. 6a, DPF3a could indeed form a complex with SNIP1, SMAD4, and p300. Then, to check whether the interaction of DPF3a with SMAD4 and p300 was dependent on SNIP1, we assayed their interaction in 786-O cells after knocking down SNIP1. As shown in Fig. 6b, the association of DPF3a with SMAD4 and p300 was notably reduced upon SNIP1 knockdown. In

**Fig. 5 | Co-regulation of cell migration-related genes by DPF3a and SNIP1.**
**a, b** The migration ability of 786-O cells after SNIP1 knockdown/overexpression as well as SNIP1 knockdown/overexpression together with DPF3a overexpression was measured using Transwell assay. Data were presented as mean values ± SD (*n* = 3 independent experiments). Statistical significance was estimated using a two-sided student *t*-test. Scale bar, 200 μm. **c, d** The metastatic ability of 786-O cells with SNIP1 knockdown and overexpression was measured using a mouse metastasis model. Tumor cells expressing luciferase were visualized with D-Luciferin as substrate using an in vivo imaging system. Total luminescence intensity was quantified and statistical significance was estimated using a two-sided student *t*-test. Data were presented as mean values ± SD (*n* = 4 animals). **e** The MA plot comparing the gene expression profile of 786-O cells with and without SNIP1 knockdown. The x-axis and y-axis represent Log2 (mean read counts) and Log2 (fold change), respectively. Red dots indicate upregulated genes, whereas blue dots indicate downregulated genes. The results

represent two independent biological replicates. **f** Genome-wide co-occupancy of DPF3a and SNIP1. Average profiles and heatmaps of normalized read density of ChIP-seq for DPF3a and SNIP1 are presented. The results represent two independent biological replicates for each ChIP-seq experiment. **g** Genome browser view of ChIP-seq and RNA-seq normalized read counts at *SNAI1*, *MMP2*, *COL1A1*, and *WNT7B* loci. Co-binding regions of DPF3a and SNIP1 are marked with a red dashed box. **h** DPF3a ChIP experiments were performed in DPF3a-OE cells after SNIP1 knockdown and DPF3a enrichment at selected genes was quantified by qPCR. Data were presented as mean values ± SD (*n* = 3 independent experiments). Statistical significance was estimated using a two-sided student *t*-test (\**p* < 0.05). $p_{COL1A1}$ = 0.029, $p_{COL6A3}$ = 0.037, $p_{SNAI1}$ = 0.046, $p_{MMP2}$ = 0.016, $p_{MMP16}$ = 0.036, $p_{WNT7B}$ = 0.023. **i** SNIP1 ChIP-qPCR was performed to compare its enrichment at selected genes in 786-O cells with and without DPF3a overexpression. Data were presented as mean values ± SD (*n* = 3 independent experiments). Source data are provided in the Source Data file.

contrast, the interaction of SNIP1 with SMAD4 and p300 stayed the same in 786-O cells with and without DPF3a overexpression (Fig. 6a). Together, these results demonstrated that SNIP1 bridged DPF3a to form a complex with SMAD4 and p300.

To further check how the formation of DPF3a-SNIP1-SMAD4-p300 complex was reflected at chromatin binding, we performed ChIP-seq using an antibody against SMAD4 and p300 on 786-O cells, respectively. We got a total of 46,367 binding sites for SMAD4, of which 21.06%, 34.57 and 44.34% were at the promoter, gene-body, and intergenic region, respectively (Supplementary Fig. 7a). For p300, 20,380 binding sites were identified, with 23.25, 26.96, and 49.76% located at the promoter, gene-body, and intergenic region, respectively (Supplementary Fig. 7b). We then correlated DPF3a-SNIP1 co-binding sites with SMAD4 binding sites and found that 92.6% of DPF3a-SNIP1 co-binding sites were bound by SMAD4, suggesting that SMAD4 recruited DPF3a and SNIP1 to specific genomic loci. Further comparison with p300 binding resulted in 6921 co-binding sites for DPF3a-SNIP1-SMAD4-p300 (Supplementary Fig. 7c), indicating a genome-wide co-localization of these four proteins (Fig. 6c). Again, we observed co-occupancy of these four proteins at promoters or intragenic regions of several cell migration associated genes (Supplementary Fig. 7d).

It has been shown that SNIP1 could suppress the activity of p300 by direct protein-protein interaction[29,35]. To clarify whether SNIP1 could directly repress p300 HAT activity in our system and, more importantly, whether the binding of DPF3a could relieve such an inhibitory effect, we used in vitro HAT assay to examine the HAT activity of the p300 core catalytic unit (a.a. 965–1810) in the presence of SNIP1 and with or without DPF3a. As shown in Supplementary Fig. 7e, the recombinant SNIP1-FL and SNIP1-IDR but not the C-terminus of SNIP1 could significantly repress the HAT activity of p300, suggesting the repressive function of SNIP1 was mainly mediated by its IDR. In contrast, we did not observe any effects of recombinant proteins of DPF3a-FL, N-terminus, and C-terminus as well as the GST control on p300 activity. Interestingly, the repressive effect of SNIP1-FL and SNIP1-IDR on p300 HAT activity could be remarkably rescued by the addition of DPF3a-FL (Fig. 6d). Furthermore, we checked whether the derepression by DPF3a required the intact full-length proteins or only part of them, and observed that both DPF3a N-terminus and C-terminus could release the inhibitory effects of SNIP1-FL and SNIP1-IDR on p300 HAT activity (Fig. 6d), consistent with the observed in vitro interaction of SNIP1 with both DPF3a N-terminus and C-terminus.

Considering the critical role of p300-mediated histone acetylation in transcription activation, we performed ChIP-qPCR using antibodies against H3ac at several DPF3a-SNIP1 co-regulated genes in 786-O cells with or without DPF3a expression. As shown in Fig. 6e, we observed a striking increase of H3ac enrichment at selected targets upregulated in both DPF3a overexpression and SNIP1 knockdown

situation (Supplementary Fig. 7f, g). In line with this, a similar increase of H3ac enrichment could also be observed on these targets upon SNIP1 knockdown (Supplementary Fig. 7h). Finally, to check whether these genes were regulated by DPF3 in the primary ccRCC cell line with a decent DPF3 expression level, we analyzed the expression as well as H3ac binding of these target genes after DPF3 knockdown. As shown in Supplementary Fig. 7i, j, we observed the decreased expression as well as reduced H3ac enrichment on them upon DPF3 knockdown. Together, these findings suggested that DPF3a activated p300 by binding to SNIP1, thereby releasing its repressive effect on p300 activity and consequently enhancing local histone acetylation and activating the transcription of target genes.

## Discussion

A previous genome-wide association study (GWAS) has implicated DPF3 as an important player in ccRCC. More recently, it was reported that DPF3 overexpression could increase growth rate in cell lines representing papillary and clear cell subtypes[24], whereas genetic knockout of DPF3 in human urinary primary tubular cells significantly decreased cell proliferation in vitro[25]. In this study, we validated the effect of DPF3 on cell proliferation both in vitro and in vivo. Moreover, we demonstrated DPF3a, the short isoform of DPF3, as an important modulator of metastasis in ccRCC and revealed the mechanism underlying DPF3a-mediated gene regulation. As summarized in Fig. 7, in a normal kidney cell with low DPF3 expression, SNIP1 represses SMADs-dependent TGF-β signaling via inhibiting p300 HAT activity, resulting in suppression of cell migration-related genes. In ccRCC patients, particularly those with *VHL* mutations, DPF3 was upregulated. Subsequently, accumulated DPF3a binds and releases the repressive effect of SNIP1 on p300 HAT activity, leading to enhanced histone acetylation that eventually activates the transcription of cell migration-related genes and consequently promotes ccRCC metastasis.

Components of the SWI/SNF complex have been reported mainly as suppressors of cell proliferation in tumor pathogenesis[11,36]. With respect to tumor metastasis, the core catalytic subunit BRG1 was found to directly interact with ZEB1, which was required for the induction of EMT in MCF7 breast cancer cells[37]. In hepatocellular carcinoma, ARID2 represses EMT by recruiting DNMT1 to suppress SNAI1[38]. In ccRCC, PBRM1 was suggested to be a potential prognostic and predictive marker in both localized and metastatic tumor[39]. However, functional evidence was missing regarding how components of the SWI/SNF complex are involved in ccRCC metastasis. Here we reported that DPF3a functioned as a metastasis activator via specifically interacting with SNIP1. This interaction was mediated by the IDR in SNIP1 and the C-terminus in DPF3a, as demonstrated by our Co-IP assay. Although the N-terminus could also bind to SNIP1 in the in vitro assay, such interaction was not observed in cells, likely because the N-terminus was occupied by other proteins such as BRG1/BRM in vivo[30,31,40]. It should be noted that the C-terminus is unique for DPF3a among the D4

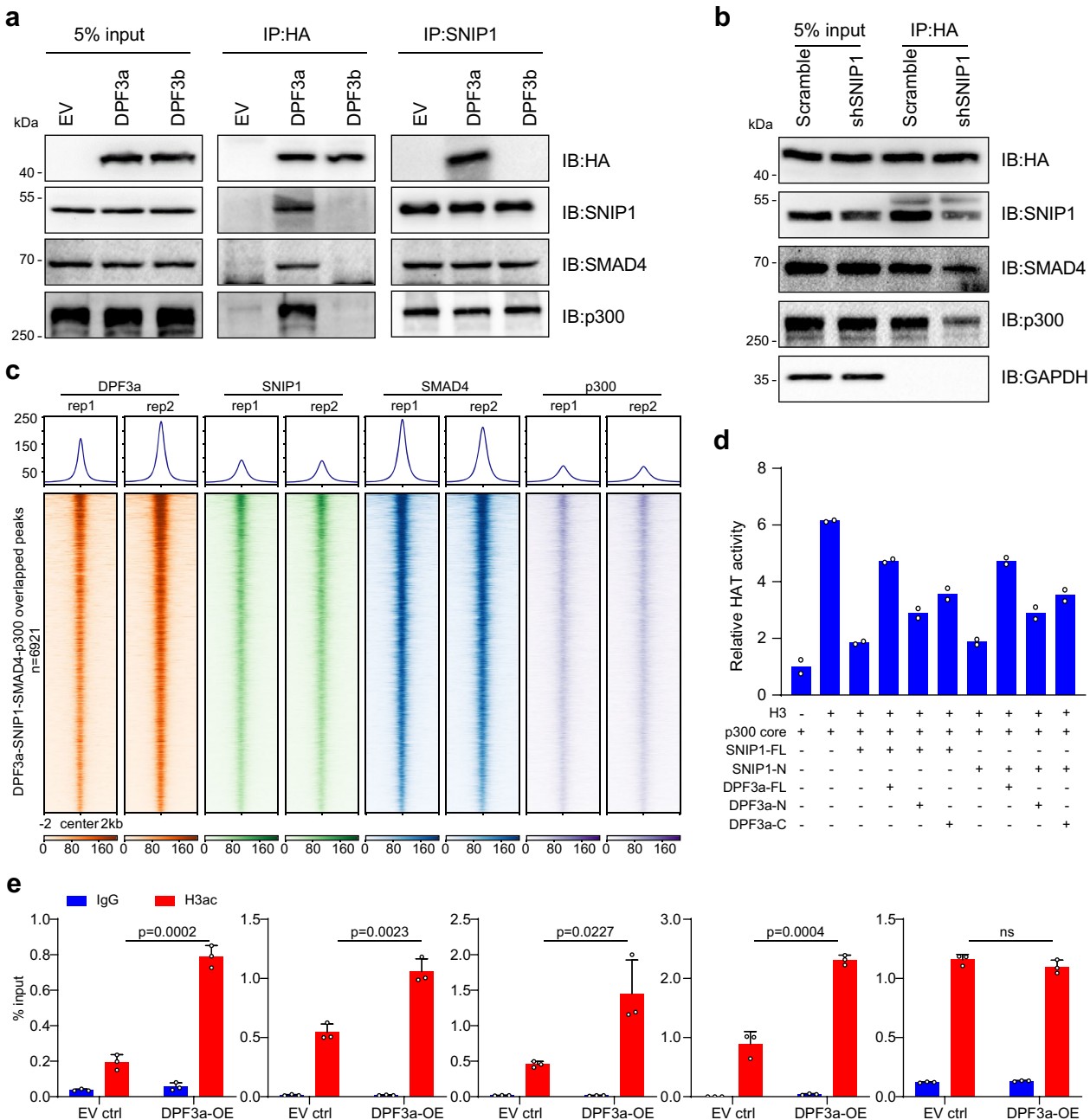

**Fig. 6 | DPF3a-SNIP1 binding released the inhibitory effect of SNIP1 on p300 activity. a** DPF3a, SNIP1, SMAD4, and p300 form in a complex. HA-tagged DPF3a or endogenous SNIP1 protein was immunoprecipitated and followed by immunoblotting using antibodies against SMAD4 and p300 to detect the co-precipitated endogenous SMAD4 and p300. Three independent experiments were performed and similar results were obtained. **b** SNIP1 bridges DPF3a to SMAD4 and p300. SNIP1 was knocked down in 786-O cells using shRNA. HA-tagged DPF3a protein was immunoprecipitated using anti-HA magnetic beads and followed by immunoblotting using antibodies against SNIP1, SMAD4, and p300 to detect the co-precipitated endogenous SNIP1, SMAD4, and p300. Input lanes represent 5% of total protein lysate. Three independent experiments were performed and similar results were obtained. **c** Co-occupancy of DPF3a, SNIP1, SMAD4, and p300 on the genome. Average profiles and heatmaps of normalized read density of ChIP-seq for DPF3a, SNIP1, p300, and SMAD4 overlapped peaks ($n = 6,921$). DPF3a and SNIP1 ChIP-seq data used in this figure are identical to the data used in Fig. 5f. **d** DPF3a releases SNIP1-mediated inhibition of p300 HAT activity in vitro. In vitro HAT activity assays of p300 in the presence of recombinant full-length SNIP1 (FL) or its N-terminus in combination with full-length DPF3a (FL), its N-terminus or C-terminus. Data were presented as mean values ($n = 2$ independent experiments). **e** Overexpression of DPF3a enhances local histone acetylation at specific loci. Pan-H3ac ChIP experiments were performed and H3ac enrichment at selected regions in DPF3a-OE cells was compared to EV control. Enrichment of H3ac was quantified by qPCR. Data were presented as mean values ± SD ($n = 3$ independent experiments). Statistical significance was estimated using a two-sided student t-test. Source data are provided in the Source Data file.

protein family, in which the C-terminus of DPF3b, DPF1, and DPF2 all contain a tandem PHD domain recognizing methylated and acetylated histone tails. This could explain why DPF3b and likely also DPF1/2 would exert no effect on cell migration in ccRCC cells.

Moreover, our in vitro HAT assay demonstrated that the binding of DPF3a to SNIP1 could relieve the inhibition of p300 HAT activity by SNIP1, which suggested the metastasis-promoting effect of DPF3a observed here was likely independent of the SWI/SNF complex activity.

**Normal**

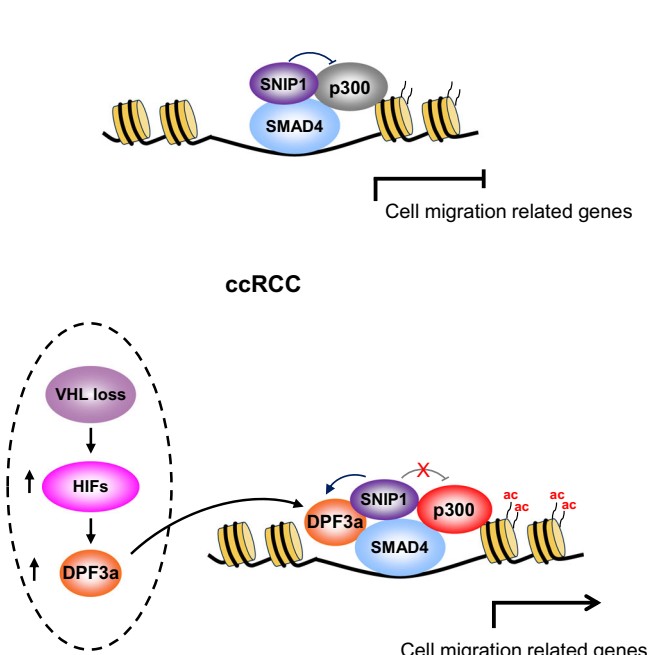

**ccRCC**

**Fig. 7 | Model and summary of the findings.** In normal conditions, SNIP1 binds to SMAD4 and inhibits the associated p300 HAT activity, which leads to the transcriptional repression of cell migration-related genes. In ccRCC patients, loss of VHL activates hypoxia-induced factors (HIFs) that may upregulate the expression of DPF3a. The latter binds to SNIP1 and releases its repressive effect on p300 activity, leading to enhanced local histone acetylation. Consequently, cell migration-related genes are activated, which eventually promote ccRCC metastasis.

However, we did identify that 67.8% of DPF3a binding sites were not co-occupied by SNIP1 and only about 12.4% of differentially expressed genes upon DPF3a-OE were dysregulated in a consistent manner after knocking down SNIP1. Therefore, the SNIP1-SMAD4-p300 axis was not the only point via which DPF3a regulated gene expression.

Depending on cell context and tumor stage, TGF-β signaling could function either as a tumor suppressor or enhancer[41]. Usually, TGF-β could inhibit cell proliferation and induce apoptosis in normal epithelium. However, cancer cells often developed an evasion of such inhibitory effects of TGF-β and took advantage of the intact TGF-β-mediated cellular responses to promote tumor progression[42]. In this study, we demonstrated that TGFB1 treatment enhanced the migration and invasion capacity of 786-O cells and DPF3a exerted the migration promoting effect via the activation of the TGF-β pathway. The link between DPF3a and TGF-β was through SNIP1, which could negatively regulate the migration of ccRCC cells via repressing TGF-β signaling. Previously, two mechanisms have been demonstrated for SNIP1-mediated inhibition of TGF-β activity. Kim et al. reported that SNIP1 could bind to the CH1 domain of p300, thereby preventing the SMAD4-p300 interaction[29], whereas Xing et al. suggested that SNIP1 interacted with PHD and CH3 domains in p300 catalytic unit to directly exert its inhibitory function[35]. In our in vitro HAT assay, we observed that SNIP1 could inhibit the HAT activity of the p300 core catalytic unit, which was in line with the second mechanism proposed before. Moreover, we found the IDR of SNIP1, which directly interacted with both DPF3a and p300, was essential for repressing the enzymatic activity of p300. It is therefore mechanistically plausible that DPF3a released the inhibitory effects of SNIP1 by competing with p300 for binding to SNIP1. In addition to its repressive role in regulating TGF-β signaling, SNIP1 was also shown to function as a transcriptional co-activator of c-MYC in DNA damage response[43,44]. However, we did not observe alterations of known MYC targets such as *BCL2*, *CCND1*, *PCNA*,

*PGK1*, and *VEGFA*[45] upon SNIP1 knockdown, and MYC DNA binding motif was not enriched in our SNIP1 ChIP-seq data. We, therefore, assumed that, at least in our system, SNIP1 mainly acted as a repressor of p300-dependent TGF-β signaling to enhance the migration capacity of ccRCC cells.

In addition to its regulatory role in transcription, SNIP1 has been linked to the regulation of splicing. Unlike its widespread effects on intron splicing during zebrafish embryogenesis[33], we only observed mild alteration of splicing upon SNIP1 knockdown in 786-O cells. This apparent inconsistency was likely due to the use of different experimental models (cancer cell vs zebrafish early embryo) or different gene perturbation methods (knockdown vs genetic knockout). Nevertheless, it is highly implausible that the effect of DPF3a on cell migration observed in our study was due to splicing dysregulation via SNIP1 because (1) very few splicing changes were commonly observed between DPF3a overexpression and SNIP1 knockdown and (2) splicing of the commonly affected genes has not been reported to be involved in cell migration.

Finally, in ccRCC patients, *VHL* mutations were frequently observed. We found that *DPF3* was significantly upregulated in ccRCC patients, especially those with *VHL* mutations. Loss of VHL leads to increased protein levels of HIFs, which in turn activates the transcription of genes involved in proliferation, apoptosis, and metastasis[7,46]. The risk locus (rs4903064) located within the first intron of *DPF3* has been recently reported to create a hypoxia-response enhancer[24,25], whose activity is regulated by VHL. The expression of DPF3 was then suggested to be modulated by the binding of HIFs in an allele-specific manner[25]. Considering the important role of the hypoxia pathway in ccRCC development and progression, it would be interesting to decipher to what extent DPF3 contributes to HIFs-mediated ccRCC pathogenesis in the future.

## Methods

### Cell culture
The 786-O (CRL-1932) and HEK293 (CRL-3216) cells were obtained from the ATCC. 786-O cells were cultured in RPMI 1640 (Gibco) with 1% penicillin/streptomycin (Gibco) and 10% FBS (Gibco). HEK293T cells were cultured in DMEM (Gibco) supplemented with 1% penicillin/streptomycin (Gibco) and 10% FBS (Gibco). The primary cells isolated from primary tumors of ccRCC patients were obtained from Dr. Walter Birchmeier's Lab at Max-Delbrueck-Center for Molecular Medicine, Berlin[26] and were cultured in EMEM (ATCC) supplemented with 10% FBS (Gibco), 1% NEAA (Gibco), 2 mM L-Glutamin (Gibco), 1% Penicillin/Streptomycin (Gibco), and 1.25 μg/ml Amphotericin B (Sigma). All cells were cultured at 5% $CO_2$ and 37 °C.

### Cloning Procedures
The expression vector encoding human *DPF3a* and *DPF3b* were constructed by replacing the Cas9 cassette on lentiCas9-Blast (Addgene) with the HA-tagged *DPF3a/DPF3b* coding sequence followed by T2A and mCherry cassettes. The expression vector encoding human *SNIP1* was constructed by replacing the Cas9 cassette on lentiCas9-puro (Addgene) with a Flag-tagged *SNIP1* coding sequence by T2A and GFP cassettes. The shRNAs targeting *SNIP1* were cloned into the PLKO.1-puro vector (Addgene). The lenti-Luciferase plasmid was constructed by replacing the Cas9 cassette on lentiCas9-Puro or lentiCas9-blast (Addgene) with a luciferase cassette. Bacterial expression vectors for *DPF3a* and *SNIP1* were constructed by subcloning the cDNA sequences into the pGEX-6P-1 vector.

### Virus package and transduction
For each virus package, HEK293T cells were seeded one day before and were transfected with the lentiviral plasmid, pMD2.G (Addgene), and PAX2 (Addgene) at a ratio of 1:1.5:1 using PEI (Sigma). The medium was refreshed 12 h after transfection. The supernatant was collected after

48 h with centrifugation and filtering through a 0.45 μm filter. The transduction was done by incubating the viral particles containing supernatant with the targeting cells overnight in the presence of polybrene (Sigma).

## RNAi knockdown

The primary ccRCC cells were seeded into a six-well plate with 2 ml media at 40% confluence. After 4 h, a mixture of 4.4 μl (20 μM) siRNA targeting *DPF3* (Qiagen) in 100 μl Opti-MEM (Thermo) and 8.8 μl lipofectamine 3000 (Invitrogen) in 100 μl Opti-MEM was incubated for 15 min at room temperature and then added to the cells. A scramble siRNA (siNon, Qiagen) was used as the negative control. After 24 h, the media was refreshed and cells were collected after 48 h to check knockdown efficiency. The sequences of siDPF3 and siNon are listed in Supplementary Table 2.

## Cell Counting Kit-8 (CCK-8) assay

Cells were seeded into the 96-well plates with $10^3$–$10^4$ cells per well. After incubation for 0, 24, 48, and 72 h, CCK-8 solution (TargetMol) was added to each well and cells were incubated for an additional 2 h. Finally, the absorbance was measured at a wavelength of 450 nm.

## TGFB1 ELISA assay

The concentrations of secreted TGFB1 in culture media were measured using a commercial ELISA kit (ab100647, Abcam) according to the manufacturer's protocol. Briefly, the cell culture medium was activated and distributed in a 96-well plate after dilution. Biotinylated TGFB1 antibody was then added to each well, followed by adding Streptavidin and Substrate solution. After adding Stop solutions, the assay was immediately measured at 450 nm.

## MMP2 activity assay

MMP2 activity was measured by InnoZyme™ Gelatinase activity assay kit (Cat. No. CBA003 Calbiochem) according to the manufacturer's protocol. Briefly, 24 h after seeding, the cell culture medium was collected and diluted in activation buffer and distributed in a 96-well plate. A substrate working solution was added to each well and the plate was incubated at 37 °C for 6 h. The fluorescence was then measured with an excitation wavelength of 320 nm and an emission wavelength of 405 nm.

## Cell migration and invasion assay

Both cell migration and invasion abilities were measured by using Transwell assays (Cat. No. 3422 and Cat. No. 354480 for migration and invasion assay, respectively, Corning). Twenty thousand cells were seeded per transwell in 120 μl RPMI 1640 medium containing 2% FBS, and RPMI 1640 medium containing 10% FBS was added to the bottom chamber. After 12 or 24 h, 4% methanal was used to fix the cells on the bottom surface for 30 min at room temperature. After staining the cells with Crystal Violet Staining Solution (Beyotime) for 20 min, the images were taken using a microscope (Nikon, #Ts2-FL) and the area covered by the cells was quantified using ImageJ.

## Immunofluorescence (IF) staining

786-O cells expressing HA-tagged DPF3a were fixed with methanol and were blocked by Immunol Staining Blocking Buffer (Beyotime) for 1 h at room temperature. Afterward, primary antibodies were applied in the blocking buffer for 2 h followed by three times washing with TBS. Secondary antibody incubation was carried out in the same buffer for 45 min followed by three times washing with TBS and DAPI staining. The cells were mounted in Immu-mount (Thermo) and examined on a confocal laser scanning microscope (Nikon A1R). The antibodies used in IF staining are listed in Supplementary Table 3.

## Animal study and in vivo imaging

Eight weeks old B-NDG female mice (stain name: NOD.CB17-Prkdc[scid] Il2rg[tm1]/Bcgen, Biocytogen) were randomized into groups ($n = 4$ for each group). All animals were housed with a 12 h light/dark cycle, temperature nominally 25 °C, and humidity 50%. For the metastasis model, $5 × 10^5$ cells infected with Lenti-Luciferase-Puro virus were suspended in 50 uL PBS and injected into the spleen using an insulin syringe. For subcutaneous inoculations, $10^6$ cells were injected into the right and left flanks of the mice. Four to eight weeks later, all the mice were injected with D-Luciferin potassium salt solution (100 ug/g, Beyotime) into the abdomen to emit luminescence from the cells injected before. After 10 min, luminescence intensity was detected using an in vivo Imaging System (PerkinElmer, #IVIS Spectrum). All mice were dissected after imaging to harvest spleens and livers for histological analysis. No blinding was done during animal experiments. The survival endpoint was when the weight loss was not exceeded 20% of the control group for metastasis models or the tumor was not exceeded 15 mm for subcutaneous models. All animal experiments were carried out following animal protocols approved by the Laboratory Animal Welfare and Ethics Committee of the Southern University of Science and Technology.

## Histological analysis

After dissection, the spleens and livers were washed using PBS three times and then fixed with a 4% formaldehyde solution. After dehydrating using a graded alcohol series (70–100%), the tissues were embedded using paraffin and cut at 6 μm thickness. The sections were stained with Hematoxylin and Eosin (HE) for histologic examination.

## Protein expression and purification

The plasmids were transformed into BL21 (DE3)-RIL cell strain (Stratagene) and cultured using a Luria-Bertani (LB) medium supplemented with proper antibiotics at 37 °C to an $OD_{600}$ of ~1.0–1.2. Then, protein expression was induced with 0.5 mM IPTG (Isopropyl β-D-Thiogalactoside) and cells were further incubated overnight at 20 °C. For the expression of DPF3a constructs, 0.1 mM ZnSO4 was supplemented together with IPTG.

The GST-tagged proteins were first purified with Glutathione Sepharose 4B beads (GE Healthcare) and were further purified by the HiLoad 16/600 Superdex 200 column (GE Healthcare). For some usages, the GST-tags of SNIP1 1–240 and full-length DPF3a were removed by 3 C protease before the final gel-filtration step. Purified protein samples were concentrated, flash-frozen in liquid nitrogen, and stored at −80 °C for further usage.

## GST pulldown

About 50 μL of Glutathione Sepharose 4B beads was suspended with 200 μL of binding buffer (20 mM Tris, pH 7.5, 100 mM NaCl, 5 mM β-mercaptomethanol), and 1.5 nmol of GST-tagged DPF3a (full length, N-terminus and C-terminus) or SNIP1 (full length, N-terminus and C-terminus) were added and incubated at 4 °C for 30 min. Then, 2 nmol of SNIP1 1–240 or full-length DPF3a proteins, respectively, were added and further incubated at 4 °C for 2 h. Then, the beads were washed four times with 1 mL of washing buffer (20 mM Tris, pH 7.5, 100 mM NaCl, 5 mM β-mercaptomethanol, 0.5% Triton X-100) before adding 50 μL of sample loading buffer. All the samples were analyzed with SDS-PAGE.

## Co-immunoprecipitation (Co-IP)

The 786-O cells and HEK293T cells were homogenized in lysis buffer containing 20 mM Tris-HCl pH 7.4, 150 nM NaCl, 1 mM EDTA, 1% Triton, 1 mM DTT, 0.1 mM PMSF, 1 mM NaVO4, protease inhibitor (Roche). After measuring protein concentrations using the BCA assay (Beyotime), the cell extracts were incubated with the indicated antibodies (Supplementary Table 3) and protein A/G beads (MCE) for 2–4 h at

4 °C. The beads were then washed three times with TBST and eluted for immunoblotting.

## Immunoprecipitation followed by mass spectrometry analysis (IP-MS)

786-O cells with and without HA-tagged DPF3a expression were washed with ice-cold PBS and homogenized in lysis buffer containing protease and phosphatase inhibitors. The DPF3a-associated proteins or SNIP1-associated proteins were immunoprecipitated with anti-HA beads (Thermo) or Protein G beads coupled with SNIP1 antibody for two hours. After three times washing with TBST, on-beads digestion with sequencing-grade trypsin (Promega) was carried out. Afterward, the samples were analyzed on an LTQ-Orbitrap Elite mass spectrometer system (Thermo). IP-MS data was processed by MaxQuant for label-free quantification with "match between run" function activated and database searching was against the UniProt human database.

## Chromatin immunoprecipitation (ChIP)

The ChIP assay was performed according to the standard protocol provided by SimpleChIP Plus Sonication Chromatin IP Kit (CST) with modifications. For DPF3a and SNIP1 ChIP, we carried out a two-step crosslink in which cells were first crosslinked with DSG (Thermo) for 45 min, followed by 1% formaldehyde fixation for 10 min at room temperature. For H3ac, SMAD4, and p300, cells were directly fixed with 1% formaldehyde for 10 min at room temperature. Afterward, sonication was carried out using the Bioruptor pico (Diagenode) by applying 10 cycles of 30 s ON and 30 s OFF to obtain chromatin fragments of approximately 100–500 bp. ChIP was performed with the indicated antibodies (Supplementary Table 3). ChIP DNA was cleaned up using the ChIP DNA Clean& Concentrator kit (Zymo).

For qPCR analysis, ChIP enrichment was normalized to the input and expressed as a relative enrichment of the material precipitated by the indicated antibody on target regions. ChIP primers are listed in Supplementary Table 4.

For ChIP-seq, libraries were prepared using a standard protocol provided by VAHTSTM Universal DNA Library Prep Kit for Illumina® V3 (Vazyme). The libraries were sequenced in a 2 × 150 nt manner on NovaSeq 6000 platform (Illumina).

## Immunoblotting

Cells were homogenized in lysis buffer (20 mM Tris-HCl pH 7.4, 150 nM NaCl, 1 mM EDTA, 1% Triton, 1 mM DTT, 0.1 mM PMSF, 1 mM NaVO4, protease inhibitor (Roche)), and then centrifuged for 15 min at 10,000×$g$ and 4 °C. The protein concentration was measured using a BCA assay (Beyotime). Immunoblotting was performed according to standard protocols. All antibodies, with their respective dilutions, are given in Supplementary Table 3.

## In vitro p300 HAT assay

The in vitro p300 HAT assay was performed using the HAT assay kit (Active Motif) according to the manufacturer's instructions. Briefly, p300 core catalytic domain (aa965–1810) was incubated with Histone H3 peptide in the presence of Acetyl-CoA. Different combinations of recombinant SNIP1 fragments and/or DPF3a fragments (see main text for the detail) were then added for 30 min at room temperature. Afterward, the reaction was stopped and developed for 15 min in dark at room temperature. The arbitrary fluorescence units were measured with excitation at 360–390 nm and emission at 450–470 nm.

## RNA extraction and expression analysis

Total RNA of cultured cells was isolated using RNA Isolator Total RNA Extraction Reagent (Vazyme), followed by isopropanol precipitation according to the standard protocols. Reverse transcription reactions were performed using the HiScript III 1st Strand cDNA Synthesis Kit (Vazyme) with random hexamers. Quantitative real-time PCR measurements were carried out using Hieff qPCR SYBR Green Master Mix (Yeasen) on the BIO-RAD real-time PCR system. Gene expression was calculated according to ΔCT method with normalization to the housekeeping gene *GAPDH* or *ACTB*. The primers used to measure the mRNA expression are listed in Supplementary Table 4.

For mRNA sequencing, libraries were prepared using the standard protocol provided by VAHTS Stranded mRNA-seq Library Prep Kit (Vazyme) with starting material of 1 μg total RNA. The libraries were sequenced in a 2 × 150 nt manner on NovaSeq 6000 platform (Illumina).

## ATAC-seq library preparation and sequencing

For ATAC library construction, cells were washed and then lysed in 50 μL lysis buffer (10 mM Tris-HCl (pH 7.4), 10 mM NaCl, 3 mM MgCl2, 0.1% NP-40, 0.1% Tween-20, and 0.01% digitonin) for 3 min on ice. Immediately after lysis, samples were then incubated with the Tn5 transposase and tagmentation buffer at 37 °C for 30 min (Vazyme Biotech, TD501). PCR was then performed to amplify the library for 12 cycles using the following PCR cycles: 72 °C for 3 min; 98 °C for 30 s, followed by thermocycling at 98 °C for 15 s, 60 °C for 30 s, and 72 °C for 40 s, and finally 5 min at 72 °C. After PCR, libraries were purified with 1.2X DNA clean beads (Vazyme Biotech, N411). The libraries were sequenced in a 2 × 150 nt manner on NovaSeq 6000 platform (Illumina).

## Clinical data analysis

Normalized RNA expression data of samples from the TCGA-KIRC cohort were extracted by TCGA-Assembler[47], and then the tumor samples were compared to the adjacent normal. TCGA-KIRC samples were split into two groups based on either VHL, PBRM1, or SMARCA4 mutations status from cBioPortal[48]. FeatureCounts was applied to quantify the expression of each exon in DPF3 by using bam files from TCGA[49]. The level of DPF3 expression in metastasis and primary tumors were obtained from HCMDB (the Human Cancer Metastasis Database)[50].

## RNA-seq analysis

For RNA-seq analysis, quality control and adapter trimming were performed using fastQC (http://www.bioinformatics.babraham.ac.uk/projects/fastqc/) (v0.11.2) and cutadapt[51] (v1.14), respectively. The RNA-seq reads were mapped to the hg38 reference genome by HISAT2[52] (version 2.1.0). Counts for known genes were generated using featureCounts[49] (version 2.0.1) with the GENCODE v34 annotation file. Gene expression was quantified using transcripts per million (TPM), and genes with low expression (TPM <2) in all samples were filtered out. To identify differentially expressed genes, the R/Bioconductor package "DESeq2"[53] was used. Normalized read coverage tracks (bigwig format) were generated for visualization using bamCoverage in the deepTools package[54], with the parameters --normalizeUsing RPKM -bs 10. The RNA-seq data were visualized by using Integrative Genomics Viewer (IGV, version 2.5.3)

To analyze the splicing patterns, the RNA-seq reads were mapped to the hg38 reference genome by STAR[55] (version 2.7.8a) with parameters --alignEndsType EndToEnd. The unique mapped reads were processed by rmats[56] (version 4.1.1) with parameters -t paired --readLength 150 --tstat 20. MA plots were generated for the five different types of splicing events, including skipped exon (SE), alternative 5′ splice site (A5SS), alternative 3′ splice site (A3SS), mutually exclusive exons, and retained intron (RI). The statistical significance was examined with a hierarchical model to simultaneously account for sampling uncertainty in individual replicates and variability among replicates. A threshold of FDR <0.05 and |ΔPSI| >0.1 were used to define differentially spliced events.

## ChIP-seq analysis

To analyze ChIP-seq data, the sequencing reads were mapped to the hg38 reference genome by Bowtie2[57] (version 2.3.5.1) after quality control and adapter trimming (same as above). The reads with MAPQ lower than 30 or mapped to mitochondria were removed for further analysis. After duplicate removal by Picard tools (http://broadinstitute.github.io/picard), peaks were called using MACS2[58] (version 2.2.5) with the parameters -g hs. Peaks with strong signals (peak integer score >50, i.e., int(−10*log10qvalue) >50) were retained for further analysis. Overlap between peak sets were performed using the BEDTools[59] intersect function requiring reciprocal overlap with a minimum fraction of 0.5 (-f 0.5 -r). Peak distribution was calculated by the HOMER tool[60] annotatePeaks.pl using the summits of the peaks. Heatmaps were generated together with the normalized read coverage tracks using deepTools[54]. The ChIP-seq data were visualized by using Integrative Genomics Viewer (IGV, version 2.5.3).

## ATAC-seq analysis

For ATAC-seq analysis, the reads were trimmed using fastp[61] with parameters -a CTGTCTCTTATA --detect_adapter_for_pe -w 12 --length_required 20 -q 30. The reads were aligned to the hg38 reference genome by Bowtie2[57] (-X 2000), and reads with MAPQ lower than 30 or mapped to mitochondria were removed. Duplicated reads were removed using sambamba v0.7.0[62]. MACS2[58] was used to call peaks with parameters -g hs --keep-dup all -q 0.05 --slocal 10000 --nomodel --nolambda -B --SPMR. For comparative analysis, reads were counted using featureCounts[49] and the counts were converted to CPM (counts per million) for plotting.

## Reporting summary

Further information on research design is available in the Nature Research Reporting Summary linked to this article.

## Data availability

The publicly available dataset from TCGA used in this study are available in Genomic Data Commons at National Cancer Institute [https://portal.gdc.cancer.gov]. All the sequencing data generated from this study have been submitted to the NCBI under the accession number GSE175848. The proteomics data have been deposited to the ProteomeXchange Consortium via the PRIDE partner repository with the dataset identifier PXD033421. The remaining data are available within the Article, Supplementary Information, or Source Data file. Source data are provided with this paper.

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

## Acknowledgements

This work was supported by the National Natural Science Foundation of China (Grant No. 31900462), Shenzhen Science and Technology Program (Grant No. KQTD20180411143432337), and Shenzhen Key Laboratory of Gene Regulation and Systems Biology (Grant No. ZDSYS20200811144002008). The authors acknowledge the SUSTech Core Research Facilities for technical support and the Center for Computational Science and Engineering of SUSTech for the support of computational resources.

## Author contributions

W.C. and H.C. conceived the project and wrote the manuscript. W.C. supervised the project. H.C. designed and performed experiments. Y.T. prepared RNA-seq libraries and contributed to the IP-MS experiments. C.T. performed RNA-seq, ChIP-seq, and ATAC-seq analysis. H.B. performed in vitro GST pulldown and purified proteins for the HAT assay. B.Z. performed the analysis of clinical data. H.Y., L.F., and Y.T. contributed to in vitro and in vivo migration assays. R.T. consulted the Mass spec experiment and analysis. X.S., D.G., W.L., R.C., and Q.Z. assisted in performing experiments. W.C., H.C., X.G., H.H., Y.H., and S.R.S. reviewed and discussed the results.

## Competing interests

The authors declare no competing interests.
