## [Peer Review File · Nature Communications]

The SWI/SNF chromatin remodeling factor DPF3 regulates metastasis of ccRCC by modulating TGF- β signalingReviewers' Comments:

Reviewer #1:

Remarks to the Author:

The authors follow up on two previous reports that have linked overexpression of DPF3 with ccRCC. They report the important discovery that the DPF3a splicing form drives cell proliferation in kidney cancer cells rather than the DPF3b variant. This is relevant because the DPF3a lacks the two PHD fingers which interact with modified histones and are potential targets for small molecules - showing that attempts to target these domains for new therapies would be fruitless.

They also analyze existing data sets to show that DPF3 gene is frequently overexpressed with kidney cancer, and this seems to be associated with mutations in VHL. Surprisingly, they do not comment on the mutational status of PBRM1, another subunit of SWI/SNF complexes that is frequently mutated in ccRCC. The discussion of how mutation or overexpression of subunits of SWI/SNF complexes relates to cancer does not take into account recent extensive recent advances in this area, and this needs to be addressed to put their results in context for readers.

They also carry out proteomics studies to try to identify a protein that specifically interacts with DPF3a, and reports a putative interaction with the protein SNIP1. Based on this they propose a mechanism for how overexpression of DPF3a disrupts the regulation of cell proliferation by controlling the activity of P300. The literature describing the interaction between SNIP1 and P300 is 20 years old, and it has not been followed-up in any detail. SNIP1, however, has been more recently linked to the regulation of splicing, and there are in fact several cryoEM structures of spliceosomal complexes that contain this protein. Even if the link with P300 is correct, SNIP1 clearly has a role in splicing, as do SWI/SNF complexes. This link needs to be discussed.

Although this is an interested area and clinically important, I do not feel that this comes together as a mechanistic study.

Reviewer #2:

Remarks to the Author:

In the current manuscript, Cui et al. reported the mechanisms underlying the role of DPF3 in modulating metastasis of ccRCC. The authors uncovered that the short isoform of DPF3, i.e., DPF3a, is overexpressed in ccRCC patients. By employing a battery of techniques, including various next-generation sequencing-based approaches, molecular biology, and animal models, the authors found that DPF3a could interact with SNIP1, and this interaction represses the interaction between SNIP1 and SMAD4-p300 histone acetyltransferase complex, thereby giving rise to elevated acetylation of core histone proteins and activation of cell movement-related genes. Together this is highly novel study, the experiments were rigorously designed and carefully conducted, and the manuscript is very well-written. I recommend the acceptance of the paper for publication in Nat. Commun. after some revisions.

Major comments:

The characterizations of the roles of DPF3 in metastatic transformation in vitro can be improved. In particular, the authors only assessed how genetic perturbation of DPF3 affects migration of cultured cells, and its effect on the invasive capacity of cancer cells was not assessed. The transwell assay can be adapted for such assessment. Likewise, while the authors showed that DPF3 positively regulates the expression of MMP2 gene, it was not examined whether this elevated expression gives rise to increased activities of secreted MMP2 and MMP9, two matrix metalloproteinases that are important in promoting the invasion of cancer cells.

In the introduction, the authors mentioned that genetic mutations of components in the SWI/SNF chromatin remodeling complex are among the most common in ccRCC. It might be worth discussing that SETD2, a histone methyltransferase catalyzing the formation of trimethylation of lysine 36 in

histone H3, is also commonly mutated in ccRCC.

The ChIP-Seq data revealed substantial co-occupancy of DPF3a and SNIP1 in promoters and intragenic regions. The authors should check their RNA-seq data to determine if genetic perturbation of DPF3a perturbs the alternative splicing of cell movement-related genes.

In the text, the authors mentioned that VHL mutation may give rise to increased expression of DPF3a. How much is known about the mechanism about this regulation?

Minor comments:

Line 201, "inhibition" can be changed to "genetic depletion".

Line 217, "consisted" should read as "consistent".

Line 289, change "genome wide" to "genome-wide".

Line 292, change "DPF3a mediated ... a normal kidney cells" to "DPF3a-mediated ... a normal kidney cell line".

Line 307, change "CoIP" to "Co-IP".

Line 324, change "SNIP1 mediated" to "SNIP1-mediated".

Page 13, the first paragraph is too long, and it can be broken down to 2-3 paragraphs.

Reviewer #3:

Remarks to the Author:

Cu et al. describe the novel mechanisms for the regulation of metastasis in ccRCC cells.

The authors focused on the role of DPF3a on the enhanced expression of targets for TGF- β that are related to cellular movement.

In the manuscript, several kinds of biochemical analysis, as well as biological experiments, were employed.

The paper's main topic is of importance and might be useful for the potential readers of the journal. However, I would like to raise the following comments and concerns.

Major

1) First of all, several figures and related legends were misplaced or missing.

2) Evaluation of metastatic phenotype of ccRCC cells is critical for the paper.

Thus, authors are required to determine whether EMT is regulated by the DPF3a and SNIP1 based on the expressions of epithelial/mesenchymal markers or morphological features.

Moreover, the effect of SNIP1 expression on metastasis of ccRCC cells must be examined in vivo (Fig. 5).

3) Although the Smad4-p300-mediated gene transcription is mainly focused, it is not evident that extracted genes are actually regulated by TGF- β .

Also, the source for TGF- β is not presented in each experiment.

In many cases, targets for TGF- β are induced in the presence of exogenously stimulated TGF- β .

Authors should directly show that the expression level of genes in Supplementary Fig. 3a is induced by TGF- β in ccRCC cells.

Then, its involvement in cancer metastasis must be addressed.

If endogenous TGF- β signaling is supposed to be activated, autocrine production of TGF- β ligands must be shown.

4) Authors indicated that DPF3a and SNIP1-mediated mechanism is important for regulating of HAT activity of several genes.

However, its involvement in the expression of metastasis-related genes or the metastatic ability of cancer cells is not directly indicated.

It is required to show that the actual expression level of specific genes in Fig. 6e is regulated by DPF3a and SNIP1 in ccRCC cells.

While I am briefly familiar with the ATAC-seq, it might be better to illustrate briefly how many percentages of Smad4 targets may be altered by DPF3a-SNIP1 in ccRCC cells.

Minor

1) Throughout the manuscript, only 786-O cells are employed for many experiments.

Authors are required to reproduce several important results by knockdown experiments using cancer cells that express DPF3a.

2) Several designs of siRNAs or shRNAs should be used to exclude the off-target effects.

3) I could not find confirmation data on successful gene overexpression or silencing, Fig. 5a (shSNIP1+DPF3a-OE), Fig. 5b (SNIP1-OE-DPF3-OE), Fig. 6d, and Supplementary Fig. 6e.

4) It might be favorable if up-regulation of DPF3 proteins is shown using histopathological examinations.

Increased expression of DPF3a is shown mainly based on the re-analysis of public datasets in the current manuscript.

5) Authors claimed that "... indicating that upregulation of DPF3 is likely associated with VHL mutations in ccRCC" on page 5.

However, indirect correlation is only presented.

6) Although the metastatic liver tumor was observed in Fig. 2c, liver metastasis is not frequent in ccRCC cases.

Metastatic ability should be assessed by lung or brain metastasis induced by tail vein injection or intracardiac injection of ccRCC cells.

7) It is not unclear whether up-regulation of DPF3a is important for renal carcinogenesis or metastatic progression.

This is because the expression level of DPF3a is already raised in primary tumor tissues as shown in Fig. 1b.

Also, the authors utilized the cancer cell inoculations into the spleen and stated that "without affecting the engraftment of the cell" on page 6.

I am worried that this procedure may not be common and inappropriate for the evaluation of renal tumor formation.

Subcutaneous inoculations, if possible orthotopic inoculations are recommended.

8) I am wondering whether the interpretation of the results of Fig. 5g is appropriate.

When cells were overexpressed with SNIP1, cell migration might not be altered even after the DPF3a knockdown.

9) Similar to many other human cancers, TGF- β is reported to act as a tumor suppressor, as well as a tumor promoter, in ccRCC.

This should be discussed in the manuscript.

Reviewer #4:

Remarks to the Author:

In this study, the authors focus on the role of the BAF complex subunit DPF3 in clear cell renal cell carcinoma. They first observe an association of higher expression levels of the isoform DPF3a with the disease, and then rely on an overexpression model in the 786-O renal cell carcinoma line. There they

show increased proliferation rates and increased metastatic potential in transwell assays and in vivo, and the relation of these changes to altered expression of gene sets related to cellular movement. Using co-immunoprecipitation and proteomics, the authors then identify the well-known interaction of DPF3a with other subunits of the BAF complex, as well as several novel interactors including with SNIP1. In co-immunoprecipitation of overexpressed or recombinant proteins, they show that this interaction is mainly mediated by the N-terminus of SNIP1 and the C-terminus of DPF3a. While SNIP1 knock-down does not affect the increased migration phenotype conferred by DPF3a overexpression, the inverse is true and SNIP1 overexpression appears to antagonize DPF3a overexpression. The authors show that this antagonism is also detectable in an assay for SNIP1-mediated repression for histone acetyltransferase activity.

In summary, the authors provide a detailed characterization of a novel interaction between DPF3a and SNIP1. Given that all these experiments are conducted in an overexpression settings, it will be important to validate these findings on endogenously expressed proteins. In terms of characterizing the DPF3a isoform, very recently another report has used a similar overexpression model to study the isoform in multiple renal cell carcinoma lines including also 786-O (Colli at up AJHG 2021). Overall, this study provides similar analyses of DPF3 overexpression on gene expression and chromatin accessibility, and shows that DPF3a partially exerts its effects partially through apoptosis and STAT3 pathways. Therefore this publication at least partially compromises novelty, and it will be important to cite, cross compare results, and put the findings around SNIP1 into context of these published data.

Major points:

1. A limitation of this manuscript is that all experiments related to mechanism are performed in an DPF3a overexpression setting, and it is unclear how the levels of overexpression relate to physiological conditions. It would be important to show that some of the effects on gene expression and chromatin structure are also conserved in the knock-down model of an DPF3 expressing renal cell carcinoma cell line the authors use in initial migration studies, and to show that the interaction with SNIP1 occurs in an endogenous context.
2. It appears that the effect the authors observe is not specific to migration, and rather mostly relates to the growth advantage conferred by DPF3a overexpression. The fold-change observed on the transwell assay (Fig. 2b) appears comparable to the one in the proliferation assay (S1f). For the SNIP1 knock-down and overexpression (Fig. 5a) only transwell assays are shown, and proliferation assays should be added to allow similar comparisons.
3. For the novel interactors of DPF3a that the authors identify in proteomics experiments, including SNIP1, it is unclear whether they are binding to free DPF3a or to BAF complexes containing DPF3a. To discriminate the two scenarios, the authors should either perform the inverse experiment and do proteomics after SNIP1 pull-down in cells with and without DPF3a overexpression, or perform size exclusion chromatography on DPF3a pull-downs and blot for BAF subunits and SNIP1.
4. Mechanistically, I do not fully understand the relevance of SNIP1 in the model. The authors show that DPF3a overexpression recruits SNIP1 to promoters and other genomic sites (Fig. 5f/g). However, DPF3a overexpression still increases migration in SNIP1 knock-down conditions (Fig. 5a), so how is SNIP1 relevant? To determine the relevance of the interaction, it would be necessary to perform ChIPs for SNIP1 (and SMAD4 and p300 and H3ac) in cells that do not overexpress DPF3a, and to perform RNA-seq studies and DPF3 ChIPs in conditions of combined DPF3a overexpression and SNIP1 knock-down. Similarly, it would be important to extend the studies on histone acetylation changes from few targeted sites (Fig 6e) to genome-wide scale and include conditions of combined DPF3a and SNIP1 perturbation.

Minor points:

5. The authors show that DPF3 transcription is upregulated in ccRCC (Fig. 1b/c). However, no data are shown for other subunits of the BAF complex. Given the high prevalence of PBRM1 mutations, it will be important to correlate DPF3 expression to mutations in the complex (rather than just VHL), and to determine whether other BAF subunits are also misregulated.
6. Supplementary figure numbering seems off, e.g. what is referred to as Suppl. Fig. 2a in the text appears to be Suppl. Fig. 1b and so forth. Also the legends to the supplementary figures 1 and 2 are switched.
7. In Fig 1c, also for the last exons (BPF3b isoform), there seems to be high expression in a lot more ccRCC samples compared to normal. The representation might be misleading because of the much higher numbers of ccRCC samples. Still, there appears a clearly distinct cohort with high DPF3 expression also in the tumor sample. The authors should investigate whether this subgroup is showing exceptionally high levels also for the other exons, and whether this subgroup is correlated to VHL and PBRM1 mutation status. Detailed statistical analysis should be provided for all comparisons.
8. In the ccRCC cell line, the authors should show effects of DPF3 knock-down not only at the RNA level (Suppl. Fig. 1b) but also on the protein level. At the same time, they should investigate how modulating the expression levels of VHL, PBRM1 and SNAP1 affects DPF3 RNA and protein levels. Finally, for the overexpression experiments (Suppl. Fig. 1e), it is important to show a Western blot not only for the HA tag but also for the endogenous DPF3 protein in order to estimate the levels of overexpression.
9. At the resolution provided, it is impossible to see differences in the livers and spleens of the animals (Fig. 2d).
10. In the gene expression study with DPF3 overexpression, the authors observe a high correlation with SMARCA4-controlled gene sets (Fig. 3c), but no changes in chromatin accessibility (Fig. 3d). Can they investigate accessibility changes directly at the promoters of these SMARCA4-controlled genes?
11. The proteomics data (Fig 4a) along with previously published data show that DPF3 is excluded from PBAF complexes. Does the overexpression system simply shift the balance between BAF and PBAF complexes, thereby phenocopying a state of PBRM1 mutation?

Point-by-point response to the referees' comments

General response to the reviewers:

We thank all the referees for their time and appreciate their highly valuable comments, which helped us to substantially improve our manuscript. During the revision, we have carefully addressed all the points raised by them, as listed below.

Reviewer #1, expert in SWI/SNF chromatin remodelling complex (Remarks to the Author):

The authors follow up on two previous reports that have linked overexpression of DPF3 with ccRCC. They report the important discovery that the DPF3a splicing form drives cell proliferation in kidney cancer cells rather than the DPF3b variant. This is relevant because the DPF3a lacks the two PHD fingers which interact with modified histones and are potential targets for small molecules - showing that attempts to target these domains for new therapies would be fruitless.

They also analyze existing data sets to show that DPF3 gene is frequently overexpressed with kidney cancer, and this seems to be associate with mutations in VHL. Surprisingly, they do not comment on the mutational status of PBRM1, another subunit of SWI/SNF complexes that is frequently mutated in ccRCC. The discussion of how mutation or overexpression of subunits of SWI/SNF complexes relates to cancer does not take into account recent extensive recent advances in this area, and this needs to be addressed to put their results in context for readers.

Answer:

As suggested by the referee, in addition to VHL, we have now analyzed the association between DPF3 expression and the mutation status of another two frequently mutated SWI/SNF subunits, PBRM1 and SMARCA4 based on the data from TCGA-KIRC cohort. As shown in Supplementary Fig. 1a, the expression of DPF3 was significantly higher only in VHL mutant group, but neither in PBRM1 nor in SMARCA4 mutant group.

We have now added the discussion about mutations of SWI/SNF complex in cancer in the Introduction section. Moreover, we also included the description of the two very recent studies of DPF3 overexpression and knockout in ccRCC cells and human urinary primary tubular cells, respectively.

They also carry out proteomics studies to try to identify a protein that specifically interacts with DPF3a, and reports a putative interaction with the protein SNIP1. Based on this they propose a mechanism for how overexpression of DPF3a disrupts the regulation of cell proliferation by controlling the activity of P300. The literature describing the interaction between SNIP1 and P300 is 20 years old, and it has not been followed-up in any detail. SNIP1, however, has been more recently linked to the regulation of splicing, and there are in fact several cryoEM structures of spliceosomal complexes that contain this protein. Even if the link with P300 is correct, SNIP1 clearly has a role in splicing, as do SWI/SNF complexes. This link needs to be discussed.

Although this is an interested area and clinically important, I do not feel that this comes together as a mechanistic study.

Answer:

Indeed, SNIP1 has been recently identified as a core member of the retention and splicing complex (RES), which also consists of the U2 snRNP-associated protein RBMX2 and the bud site-selection protein BUD13. It has been shown that genetic loss of the three RES components in zebrafish caused neurodevelopmental disorders and cell death. The defect was mainly caused via widespread intron mis-splicing, with 74-79% and 7-9% of introns showed increased and decreased retention in the RES mutants, respectively. To check whether our SNIP1 perturbation in 786-O cells could result in a similar splicing defect, we compared alternative splicing patterns between SNIP1 knockdown and control samples and identified a total of 724 differentially spliced events, including 529 SE, 65 A5SS, 48 A3SS, 40 MXE and 42 RI (Supplementary Fig. 6a). In contrast to previous study in zebrafish early embryos, we did not observe a widespread increase of intron retention after SNIP1 knockdown in 786-O cells. In parallel, we also analyzed alternative splicing changes caused by DPF3a overexpression and identified 2,138 events, including 1,476 SE, 155 A5SS, 203 A3SS, 108 MXE and 196 RI (Supplementary Fig. 6b). We then overlapped the alternative splicing changes in SNIP1 knockdown with those in DPF3a overexpression (Supplementary Fig. 6c) and found only 29 common events from 26 genes (Supplementary Table. 3). None of these genes were known to be related to cell migration. Therefore, the observed effect of SNIP1 and DPF3a on cell migration was mainly through transcriptional regulation instead of splicing.

We have now added this part into Result as well as Discussion section.

Reviewer #2, expert in RNA-seq and ChIP-seq (Remarks to the Author):

In the current manuscript, Cui et al. reported the mechanisms underlying the role of DPF3 in modulating metastasis of ccRCC. The authors uncovered that the short isoform of DPF3, i.e., DPF3a, is overexpressed in ccRCC patients. By employing a battery of techniques, including various next-generation sequencing-based approaches, molecular biology, and animal models, the authors found that DPF3a could interact with SNIP1, and this interaction represses the interaction between SNIP1 and SMAD4-p300 histone acetyltransferase complex, thereby giving rise to elevated acetylation of core histone proteins and activation of cell movement-related genes. Together this is highly novel study, the experiments were rigorously designed and carefully conducted, and the manuscript is very well-written. I recommend the acceptance of the paper for publication in Nat. Commun. after some revisions.

Major comments:

The characterizations of the roles of DPF3 in metastatic transformation in vitro can be improved. In particular, the authors only assessed how genetic perturbation of DPF3 affects migration of cultured cells, and its effect on the invasive capacity of cancer cells was not assessed. The transwell assay can be adapted for such assessment. Likewise, while the authors showed that DPF3 positively regulates the expression of MMP2 gene, it was not examined whether this elevated expression gives rise to increased activities of secreted MMP2 and MMP9, two matrix metalloproteinases that are important in promoting the invasion of cancer cells.

Answer:

As suggested by the referee, we have now examined the effect of DPF3 perturbation on the invasive capacity of ccRCC cells using Transwell assay. As shown in Fig. 2a&b, genetic depletion of DPF3 could significantly inhibit the cell invasion whereas overexpression of DPF3a but not DPF3b had the opposite effect. For MMP2 activity analysis, we have carried out a fluorometric MMP2 enzymatic activity assay and observed increased activity of secreted MMP2 in DPF3a-OE cells (Fig. 3d).

In the introduction, the authors mentioned that genetic mutations of components in the SWI/SNF chromatin remodeling complex are among the most common in ccRCC. It might be worth discussing that SETD2, a histone methyltransferase catalyzing the formation of trimethylation of lysine 36 in histone H3, is also commonly mutated in ccRCC.

Answer:

As suggested by the referee, we have now included the description of additional epigenetic regulator often mutated in ccRCC, SETD2 and BAP1 in the introduction part.

The ChIP-Seq data revealed substantial co-occupancy of DPF3a and SNIP1 in promoters and intragenic regions. The authors should check their RNA-seq data to determine if genetic perturbation of DPF3a perturbs the alternative splicing of cell movement-related genes.

Answer:

This point has been also raised by referee 1. Indeed, SNIP1 has been recently identified as a core member of the retention and splicing complex (RES), which also consists of the U2 snRNP-associated protein RBMX2 and the bud site-selection protein BUD13. It has been shown that genetic loss of the three RES components in zebrafish caused neurodevelopmental disorders and cell death. The defect was mainly caused via widespread intron mis-splicing, with 74-79% and 7-9% of introns showed increased and decreased retention in the RES mutants, respectively. To check whether our SNIP1 perturbation in 786-O cells could result in a similar splicing defect, we compared alternative splicing patterns between SNIP1 knockdown and control samples and identified a total of 724 differentially spliced events, including 529 SE, 65 A5SS, 48 A3SS, 40 MXE and 42 RI (Supplementary Fig. 6a). In contrast to previous study in zebrafish early embryos, we did not observe a widespread increase of intron retention after SNIP1 knockdown in 786-O cells. In parallel, we also analyzed alternative splicing changes caused by DPF3a overexpression and identified 2,138 events, including 1,476 SE, 155 A5SS, 203 A3SS, 108 MXE and 196 RI (Supplementary Fig. 6b). We then overlapped the alternative splicing changes in SNIP1 knockdown with those in DPF3a overexpression (Supplementary Fig. 6c) and found only 29 common events from 26 genes (Supplementary Table. 3). None of these genes were known to be related to cell migration. Therefore,

the observed effect of SNIP1 and DPF3a on cell migration was mainly through transcriptional regulation instead of splicing.

We have now added this part into Result as well as Discussion section.

In the text, the authors mentioned that VHL mutation may give rise to increased expression of DPF3a. How much is known about the mechanism about this regulation?

Answer:

In our study, we found that DPF3 expression was correlated with VHL mutation status in the TCGA-KIRC cohort. As to the mechanism about this regulation, two recent publications (Colli et al., 2021, AJHG; Portze et al., 2022, JBC) reported that the SNP rs4903064 located within the first intron of DPF3, which has been associated with increased risk of ccRCC and increased expression of DPF3, could create a hypoxia response enhancer, whose activity is regulated by VHL. The expression of DPF3 was suggested to be modulated in an allele-specific manner via HIF-VHL axis. We added this to the end of the Discussion part.

Minor comments:

Line 201, “inhibition” can be changed to “genetic depletion”.

Line 217, “consisted” should read as “consistent”.

Line 289, change “genome wide” to “genome-wide”.

Line 292, change “DPF3a mediated a normal kidney cells” to “DPF3a-mediated ... a normal kidney cell line”.

Line 307, change “CoIP” to “Co-IP”.

Line 324, change “SNIP1 mediated” to “SNIP1-mediated”.

Page 13, the first paragraph is too long, and it can be broken down to 2-3 paragraphs.

Answer:

Thanks for carefully reading through our manuscript. We have corrected them accordingly in the revised manuscript.

Reviewer #3, expert in ccRCC metastatic models (Remarks to the Author):

Cui et al. describe the novel mechanisms for the regulation of metastasis in ccRCC cells. The authors focused on the role of DPF3a on the enhanced expression of targets for TGF- β that are related to cellular movement. In the manuscript, several kinds of biochemical analysis, as well as biological experiments, were employed. The paper's main topic is of importance and might be useful for the potential readers of the journal. However, I would like to raise the following comments and concerns.

Major

1) First of all, several figures and related legends were misplaced or missing.

Answer:

Thanks for carefully reading through our manuscript and we apologize for careless mistakes (Supplementary Fig. 1 and Supplementary Fig. 2 were switched). We have now corrected them in the revised manuscript.

2) Evaluation of metastatic phenotype of ccRCC cells is critical for the paper. Thus, authors are required to determine whether EMT is regulated by the DPF3a and SNIP1 based on the expressions of epithelial/mesenchymal markers or morphological features. Moreover, the effect of SNIP1 expression on metastasis of ccRCC cells must be examined *in vivo* (Fig. 5).

Answer:

As suggested by the referee, we have carried out immunoblotting on key EMT transcription factors SNAI1 and ZEB1 as well as the matrix metalloproteinase MMP2 in 786-O cells with and without DPF3a-OE (Fig. 3c). In addition, we also measured the secreted MMP2 activity (Fig. 3d).

To examine the effect of SNIP1 perturbation on cell migration *in vivo*, we have applied the same spleen-liver metastasis model and observed a significant increase of metastasis upon SNIP1 knockdown (Fig. 5c) and a slight decrease (but not statistically significant) upon SNIP1 overexpression (Fig. 5d).

3) Although the Smad4-p300-mediated gene transcription is mainly focused, it is not evident that extracted genes are actually regulated by TGF- β . Also, the source for TGF- β is not presented in each experiment. In many cases, targets for TGF- β are induced in the presence of exogenously stimulated TGF- β . Authors should directly show that the expression level of genes in Supplementary Fig. 3a is induced by TGF- β in ccRCC cells. Then, its involvement in cancer metastasis must be addressed. If endogenous TGF- β signaling is supposed to be activated, autocrine production of TGF- β ligands must be shown.

Answer:

We have now performed TGF β 1 treatment in 786-O cells by adding recombinant human TGF β 1 to a final concentration of 20 ng/ml and analyzed the expression of DPF3a target genes after 24 hours. It turned out that TGF β 1 treatment could promote cell migration/invasion (Fig. 3f) and upregulated the expression of cell movement related genes, which were also upregulated by DPF3a-OE (Supplementary Fig. 3b).

To analyze whether the endogenous production of TGF- β ligand was activated upon DPF3a overexpression, we have performed ELISA assay and found no alterations of TGF- β ligand in 786-O cells with and without DPF3a overexpression (Supplementary Fig. 3c), suggesting no autocrine regulation of TGF- β ligands in our system.

4) Authors indicated that DPF3a and SNIP1-mediated mechanism is important for regulating of HAT activity of several genes. However, its involvement in the expression of metastasis-related genes or the metastatic ability of cancer cells is not directly indicated. It is required to show that the actual expression level of specific genes in Fig. 6e is regulated by DPF3a and SNIP1 in ccRCC cells. While I am briefly familiar with the ATAC-seq, it might be better to illustrate briefly how many percentages of Smad4 targets may be altered by DPF3a-SNIP1 in ccRCC cells.

Answer:

As requested by the referee, we have now analyzed the expression level of the four genes in Fig. 6e

and the results were shown in Supplementary Fig. 7f&g.

To analyze ATAC-seq signal on SMAD4 targets, we defined the potential SMAD4 target genes as those with SMAD4 binding sites within ± 2 kb of their gene bodies based on our SMAD4 ChIP-seq data. We then compared ATAC-seq signals at these target genes between DPF3a-OE and control cells and observed no significant difference (shown below), indicating that DPF3a did not alter the chromatin accessibility of SMAD4 targets.

Figure legend: Scatter plot comparing ATAC-seq normalized read counts (CPM: counts per million) at SMAD4 targets in 786-O cells with versus those without DPF3a overexpression. Pearson's correlation test was performed. The blue and red dash line represents fold change of 1.5 and 2 respectively.

Minor

1) Throughout the manuscript, only 786-O cells are employed for many experiments. Authors are required to reproduce several important results by knockdown experiments using cancer cells that express DPF3a.

Answer:

We have examined cell invasion ability after knockdown DPF3 by using siRNA in the primary ccRCC cells which express DPF3 (Fig. 2a). Moreover, we observed decreased expression of DPF3a

targets (Supplementary Fig. 7i) as well as reduced H3ac enrichment on these targets in DPF3 knockdown situation (Supplementary Fig. 7j).

As suggested by the referee, we also tried to search for other commonly used ccRCC cell lines with a decent DPF3 expression level by screening the Cancer Cell Line Encyclopedia (CCLE) database. Unfortunately, all the commonly used ccRCC cell lines have a very low expression of DPF3 (TPM<0.5) (shown below), which are unsuitable for the suggested knockdown experiments. During the submission of our manuscript, Colli et al. reported the positive effect of DPF3 on cell proliferation of ccRCC cells by applying only the overexpression strategy in their study (Colli et al., 2021, AJHG). More recently, Portze et al. showed that DPF3 knockout in human urinary primary tubular cells, in which DPF3 was expressed, had an opposite effect on cell proliferation *in vitro* (Protze et al., 2022, JBC). In line with this, we also observed decent DPF3 expression only in one primary ccRCC cell derived from patients. It is likely that DPF3 was only highly expressed in primary cells but not in the established ccRCC cell lines. Indeed, we also noticed that the expression level of DPF3 was decreased after many passages of our primary ccRCC cells.

Table: TPM (Transcript Per Million) values of DPF3 in ccRCC cell lines.

Cell line	TPM
786O	0.070389
A498	0.042644
CAKI1	0.333424
CAKI2	0.111031
KMRC1	0.495695
KMRC2	0.189034
KMRC20	0.201634
KMRC3	0.333424
SLR21	0.070389
SLR23	0.163499
SLR24	0.31034
SLR25	0.411426
SLR26	0.432959
UOK101	0.443607

2) Several designs of siRNAs or shRNAs should be used to exclude the off-target effects.

Answer:

For DPF3 knockdown, we have used a pool of 4 siRNAs whose specificity have been validated in our previous publications (Cui et al., 2016, NAR, Lange et al., 2008, Genes Dev).

For SNIP1 knockdown, we have now added another shRNA (Supplementary Table 4) and performed knockdown experiments (Supplementary Fig. 5a) followed by migration assay (Supplementary Fig. 5c) in 786-O cells.

3) I could not find confirmation data on successful gene overexpression or silencing, Fig. 5a (shSNIP1+DPF3a-OE), Fig. 5b (SNIP1-OE-DPF3-OE), Fig. 6d, and Supplementary Fig. 6e.

Answer:

For Fig. 5a (shSNIP1+DPF3a-OE) and Fig. 5b (SNIP1-OE-DPF3-OE), we have confirmed the successful overexpression or silencing using immunoblotting (Supplementary Fig. 5b).

For Fig. 6d, and Supplementary Fig. 7e (revised version), recombinant H3 and p300 protein used *in vitro* HAT assay were provided by the HAT assay kit (Active Motif), while purified DPF3a and SNIP1 protein samples were generated by us and have been already confirmed in Supplementary Fig. 4a&f.

4) It might be favorable if up-regulation of DPF3 proteins is shown using histopathological examinations. Increased expression of DPF3a is shown mainly based on the re-analysis of public datasets in the current manuscript.

Answer:

Thanks for the suggestion. We agree that it would be ideal if we could examine the protein level of DPF3 in pathological samples. Unfortunately, we did not have access to such valuable clinical resources.

5) Authors claimed that "... indicating that upregulation of DPF3 is likely associated with VHL mutations in ccRCC" on page 5. However, indirect correlation is only presented.

Answer:

In our study, we found that DPF3 expression was correlated with VHL mutation status in the TCGA-KIRC cohort. As to the mechanism about this regulation, two recent publications (Colli et al., 2021, AJHG; Portze et al., 2022, JBC) reported that the SNP rs4903064 located within the first intron of DPF3, which has been associated with increased risk of ccRCC and increased expression of DPF3, could create a hypoxia response enhancer, whose activity is regulated by VHL. The expression of DPF3 was suggested to be modulated in an allele-specific manner via HIF-VHL axis. We added this to the end of the Discussion part.

6) Although the metastatic liver tumor was observed in Fig. 2c, liver metastasis is not frequent in ccRCC cases. Metastatic ability should be assessed by lung or brain metastasis induced by tail vein injection or intracardiac injection of ccRCC cells.

Answer:

As suggested by the referee, we have tried tail vein injection of different amount (5×10^6 and 1×10^7) of 786-O cells in B-NDG scid mice as well as B-NDG nude mice and analyzed the metastasis 2 to 6 weeks after injection. Unfortunately, we did not observe any metastasis over the mouse body (shown below). It was likely that 786-O cells were not ideal for this model as other cells such as A549 and HEPG2 worked well with us using this model. In addition, we believe that our study mainly focused on the effect of DPF3 on cell migration and the target organ might not be the most relevant point here.

Figure legend: The metastatic ability of 786-O cells was measured using a mouse metastasis model by tail vein injection in B-NDG scid mice (left) and B-NDG nude mice (right). Tumor cells expressing luciferase were visualized with D-Luciferin as substrate using an *in vivo* imaging system and we did not observe any metastasis.

7) It is not unclear whether up-regulation of DPF3a is important for renal carcinogenesis or metastatic progression. This is because the expression level of DPF3a is already raised in primary tumor tissues as shown in Fig. 1b. Also, the authors utilized the cancer cell inoculations into the spleen and stated that "without affecting the engraftment of the cell" on page 6. I am worried that this procedure may not be common and inappropriate for the evaluation of renal tumor formation. Subcutaneous inoculations, if possible orthotropic inoculations are recommended.

Answer:

Yes, we agreed that it was inappropriate to state that DPF3a-OE did not affect the engraftment of the cell. The sentence was removed in the revised manuscript.

As suggested by the referee, we have now examined the *in vivo* effect of DPF3a on tumor growth using a subcutaneous xenotransplanted model. As shown in Supplementary Fig. 2g&h, we found a positive effect of DPF3a on tumor growth *in vivo*. The effects observed in this study both *in vitro* and *in vivo* were consistent with recent observations (Colli et al., 2021, AJHG; Protze et al., 2022,

JBC). In comparison to proliferation, we felt that the effect on cell migration was much stronger as we observed enhanced migration abilities at 12 to 24 hours after seeding, whereas the proliferation difference was observed at 72 hours after seeding. Therefore, we mainly focused on cell migration in our manuscript. Now in revised manuscript, we have clarified the effect of DPF3a in cell growth and migration separately in the Result part.

8) I am wondering whether the interpretation of the results of Fig. 5g is appropriate.

When cells were overexpressed with SNIP1, cell migration might not be altered even after the DPF3a knockdown.

Answer:

The referee should have referred to Fig. 5a or Fig. 5b. We felt that we might have not presented our data in a logic manner. Now to avoid potential misunderstanding, we have reordered Fig. 5a and Fig. 5b. In Fig. 5a, we showed that DPF3a overexpression could partially rescue the migration ability inhibited by SNIP1 overexpression. In Fig. 5b, we did NOT observe further increase of migration capacity when we overexpressed DPF3a in SNIP1 knockdown cells, which indicated that DPF3a overexpression worked via SNIP1. These together suggested that the effect of DPF3a overexpression is via relieving the inhibitory effect of SNIP1.

9) Similar to many other human cancers, TGF- β is reported to act as a tumor suppressor, as well as a tumor promoter, in ccRCC.

This should be discussed in the manuscript.

Answer:

As suggested by the referee, we have now discussed the role of TGF- β in the Discussion part.

Reviewer #4, expert in SWI/SNF chromatin remodelling complex and ATAC-seq (Remarks to the Author):

In this study, the authors focus on the role of the BAF complex subunit DPF3 in clear cell renal cell carcinoma. They first observe an association of higher expression levels of the isoform DPF3a with

the disease, and then rely on an overexpression model in the 786-O renal cell carcinoma line. There they show increased proliferation rates and increased metastatic potential in transwell assays and *in vivo*, and the relation of these changes to altered expression of gene sets related to cellular movement. Using co-immunoprecipitation and proteomics, the authors then identify the well-known interaction of DPF3a with other subunits of the BAF complex, as well as several novel interactors including with SNIP1. In co-immunoprecipitation of overexpressed or recombinant proteins, they show that this interaction is mainly mediated by the N-terminus of SNIP1 and the C-terminus of DPF3a. While SNIP1 knock-down does not affect the increased migration phenotype conferred by DPF3a overexpression, the inverse is true and SNIP1 overexpression appears to antagonize DPF3a overexpression. The authors show that this antagonism is also detectable in an assay for SNIP1-mediated repression for histone acetyltransferase activity.

In summary, the authors provide a detailed characterization of a novel interaction between DPF3a and SNIP1. Given that all these experiments are conducted in an overexpression settings, it will be important to validate these findings on endogenously expressed proteins. In terms of characterizing the DPF3a isoform, very recently another report has used a similar overexpression model to study the isoform in multiple renal cell carcinoma lines including also 786-O (Colli et al. 2021). Overall, this study provides similar analyses of DPF3 overexpression on gene expression and chromatin accessibility, and shows that DPF3a partially exerts its effects partially through apoptosis and STAT3 pathways. Therefore this publication at least partially compromises novelty, and it will be important to cite, cross compare results, and put the findings around SNIP1 into context of these published data.

Answer:

During the submission of our manuscript, Colli et al. reported the positive effect of DPF3 on cell proliferation of ccRCC cells by applying only the overexpression strategy in their study (Colli et al., 2021, AJHG). More recently, Portze et al. showed that DPF3 knockout in human urinary primary tubular cells, in which DPF3 was expressed, had an opposite effect on cell proliferation *in vitro* (Protze et al., 2022, JBC). In our study, we have validated the effect of DPF3 on cell proliferation both *in vitro* and *in vivo*. However, we found that the effect on cell migration was more prominent,

which has not been explored before, and therefore focused our mechanistical investigation on cell migration. We have now cited both publications in the Introduction as well as Discussion part.

Major points:

1. A limitation of this manuscript is that all experiments related to mechanism are performed in an DPF3a overexpression setting, and it is unclear how the levels of overexpression relate to physiological conditions. It would be important to show that some of the effects on gene expression and chromatin structure are also conserved in the knock-down model of an DPF3 expressing renal cell carcinoma cell line the authors use in initial migration studies, and to show that the interaction with SNIP1 occurs in an endogenous context.

Answer:

As suggested by the referee, we have now examined cell invasion ability after knockdown DPF3 by using siRNA in the primary ccRCC cells which express DPF3 (Fig. 2a). Moreover, we observed decreased expression of DPF3a targets (Supplementary Fig. 7i) as well as reduced H3ac enrichment on these targets in DPF3 knockdown situation (Supplementary Fig. 7j).

We also tried to search for other commonly used ccRCC cell lines with a decent DPF3 expression level by screening the Cancer Cell Line Encyclopedia (CCLE) database. Unfortunately, all the commonly used ccRCC cell lines have a very low expression of DPF3 (TPM<0.5) (shown below), which are unsuitable for the suggested knockdown experiments. During the submission of our manuscript, Colli et al. reported the positive effect of DPF3 on cell proliferation of ccRCC cells by applying only the overexpression strategy in their study (Colli et al., 2021, AJHG). More recently, Portze et al. showed that DPF3 knockout in human urinary primary tubular cells, in which DPF3 was expressed, had an opposite effect on cell proliferation *in vitro* (Portze et al., 2022, JBC). In line with this, we also observed decent DPF3 expression only in one primary ccRCC cell derived from patients. It is likely that DPF3 was only highly expressed in primary cells but not in the established ccRCC cell lines. Indeed, we also noticed that the expression level of DPF3 was decreased after several passages of our primary ccRCC cells.

Table: TPM (Transcript Per Million) values of DPF3 in ccRCC cell lines.

Cell line	TPM
786O	0.070389
A498	0.042644
CAKI1	0.333424
CAKI2	0.111031
KMRC1	0.495695
KMRC2	0.189034
KMRC20	0.201634
KMRC3	0.333424
SLR21	0.070389
SLR23	0.163499
SLR24	0.31034
SLR25	0.411426
SLR26	0.432959
UOK101	0.443607

Due to the lack of a working DPF3 antibody, we have not analyzed the endogenous interaction between DPF3a and SNIP1. Actually, we have tested several commercial DPF3 antibodies and none of them worked with enough specificity. So far, the only antibody worked in our hand was a phospho-DPF3a antibody recognizing S348 phosphorylation in cardiac and muscle tissue (Cui et al., 2016, NAR). We also tested this antibody on the primary ccRCC cells with decent DPF3 expression as well as 786-O cells with DPF3a overexpression, but found that S348 of DPF3a was not phosphorylated in ccRCC cells.

2. It appears that the effect the authors observe is not specific to migration, and rather mostly relates to the growth advantage conferred by DPF3a overexpression. The fold-change observed on the transwell assay (Fig. 2b) appears comparable to the one in the proliferation assay (S1f). For the SNIP1 knock-down and overexpression (Fig. 5a) only transwell assays are shown, and proliferation assays should be added to allow similar comparisons.

Answer:

We have demonstrated that DPF3a promoted both proliferation and migration. However, its impact on cell migration was much stronger, as we could already observe enhanced migration/invasion

capacity at 12 to 24 hours after seeding. In the proliferation assay, we observed a similar growth-promoting effects only at 72 hours after seeding. Therefore, we focused on cell migration in our study.

For SNIP1 knockdown and overexpression, we have examined cell proliferation using CCK8 assay and observed no significant alterations (Supplementary Fig. 5d).

3. For the novel interactors of DPF3a that the authors identify in proteomics experiments, including SNIP1, it is unclear whether they are binding to free DPF3a or to BAF complexes containing DPF3a. To discriminate the two scenarios, the authors should either perform the inverse experiment and do proteomics after SNIP1 pull-down in cells with and without DPF3a overexpression, or perform size exclusion chromatography on DPF3a pull-downs and blot for BAF subunits and SNIP1.

Answer:

As suggested by the referee, we have now performed SNIP1 IP-MS in 786-O cells with and without DPF3a overexpression and found that overexpression of DPF3a strikingly enhanced the affinity of SNIP1 to the SWI/SNF complex (Supplementary Fig. 4b). This suggested that SNIP1 was associated with SWI/SNF complex via binding to DPF3a *in vivo*.

4. Mechanistically, I do not fully understand the relevance of SNIP1 in the model. The authors show that DPF3a overexpression recruits SNIP1 to promoters and other genomic sites (Fig. 5f/g). However, DPF3a overexpression still increases migration in SNIP1 knock-down conditions (Fig. 5a), so how is SNIP1 relevant? To determine the relevance of the interaction, it would be necessary to perform ChIPs for SNIP1 (and SMAD4 and p300 and H3ac) in cells that do not overexpress DPF3a, and to perform RNA-seq studies and DPF3 ChIPs in conditions of combined DPF3a overexpression and SNIP1 knock-down. Similarly, it would be important to extend the studies on histone acetylation changes from few targeted sites (Fig 6e) to genome-wide scale and include conditions of combined DPF3a and SNIP1 perturbation.

Answer:

We felt that we might have not presented our data in a logic manner. Now to avoid potential

misunderstanding, we have reordered Fig. 5a and Fig. 5b. In Fig. 5a, we showed that DPF3a overexpression could partially rescue the migration ability inhibited by SNIP1 overexpression. In Fig. 5b, we did NOT observe further increase of migration capacity when we overexpressed DPF3a in SNIP1 knockdown cells, which indicated that DPF3a overexpression worked via SNIP1. To demonstrate that it is SNIP1 that recruited DPF3a to its binding sites, not vice versa, we performed DPF3a ChIP in the cells with or without SNIP1 knockdown. As shown in Fig. 5h, we found a significant reduction of DPF3a enrichment at selected DPF3a-SNIP1 targets. In addition, we performed SNIP1 ChIP in cells with or without DPF3a overexpression. As shown in Fig. 5i, in contrast to Fig. 5h, we observed no significant changes of SNIP1 binding at these sites. These validated our working hypothesis that SNIP1 recruited DPF3a to their common binding sites. Moreover, using the *in vitro* biochemical assay, we have shown that DPF3a indeed released the inhibitory effect of SNIP1 on p300 activity (Fig. 6d). All these data together suggested that DPF3a was recruited by SNIP1 to DNA targets and the interaction of DPF3a to SNIP1 is important for releasing the inhibitory effect of SNIP1.

Minor points:

5. The authors show that DPF3 transcription is upregulated in ccRCC (Fig. 1b/c). However, no data are shown for other subunits of the BAF complex. Given the high prevalence of PBRM1 mutations, it will be important to correlate DPF3 expression to mutations in the complex (rather than just VHL), and to determine whether other BAF subunits are also misregulated.

Answer:

We have now analyzed the association between DPF3 expression and the mutation status of another two frequently mutated SWI/SNF subunits, PBRM1 and SMARCA4 based on the data from TCGA-KIRC cohort. As shown in Supplementary Fig. 1a, the expression of DPF3 was significantly higher only in VHL mutant group but neither in PBRM1 nor in SMARCA4 mutant group.

6. Supplementary figure numbering seems off, e.g. what is referred to as Suppl. Fig. 2a in the text appears to be Suppl. Fig. 1b and so forth. Also the legends to the supplementary figures 1 and 2 are

switched.

Answer:

Thanks for carefully reading through our manuscript and we apologize for careless mistakes (Supplementary Fig. 1 and Supplementary Fig. 2 were switched). We have now corrected them in the revised manuscript.

7. In Fig 1c, also for the last exons (DPF3b isoform), there seems to be high expression in a lot more ccRCC samples compared to normal. The representation might be misleading because of the much higher numbers of ccRCC samples. Still, there appears a clearly distinct cohort with high DPF3 expression also in the tumor sample. The authors should investigate whether this subgroup is showing exceptionally high levels also for the other exons, and whether this subgroup is correlated to VHL and PBRM1 mutation status. Detailed statistical analysis should be provided for all comparisons.

Answer:

As suggested by the referee, we have now separated the TCGA-KIRC patients into high and low groups based on the expression of exon 12, which is specific to DPF3b. This resulted in a subgroup (n=21) with high expression of exon 12 (RSEM>1). Then, we compared the expression across exons among the three groups. As shown below, we found that this high group showed high expression levels also for other exons and DPF3a was still the main isoform expressed in this subgroup. Furthermore, we analyzed the mutation status of VHL and PBRM1 in the high and low groups, respectively. We found that the high group was not correlated to VHL or PBRM1 mutation status as compared to the low group.

Figure legend: (A) Expression analysis of each exon of DPF3 (exon 6 to exon 12) in normal kidney tissues (n=72) and ccRCC patients with high (n=21) and low (n=513) DPF3 exon 12 expression. Y axis represents the expression values of expectation-maximization (RSEM). (B) Comparison of VHL and PBRM1 mutation status in high and low group. Statistical significance was estimated using the Chi-square test.

8. In the ccRCC cell line, the authors should show effects of DPF3 knock-down not only at the RNA level (Suppl. Fig. 1b) but also on the protein level. At the same time, they should investigate how modulating the expression levels of VHL, PBRM1 and SNAP1 affects DPF3 RNA and protein levels. Finally, for the overexpression experiments (Suppl. Fig. 1e), it is important to show a Western blot not only for the HA tag but also for the endogenous DPF3 protein in order to estimate the levels of overexpression.

Answer:

As explained before, we do not have a working DPF3 antibody to check the protein level of endogenous DPF3.

To examine whether modulating of VHL, PBRM1 and SNIP1 could affect DPF3 expression, we performed qPCR to check the RNA level of DPF3 after VHL re-expression, PBRM1 knockdown as well as SNIP1 knockdown or overexpression in 786-O cells. However, we did not observe alteration of DPF3 expression at RNA level in our system (shown below).

Figure legend: (A) VHL expression at protein level and DPF3 expression at RNA level after VHL re-expression in 786-O cell. (B) DPF3 expression at RNA level after PBRM1 knockdown in 786-O cell. (C) DPF3 expression at RNA level after SNIP1 knockdown and overexpression in 786-O cell.

9. At the resolution provided, it is impossible to see differences in the livers and spleens of the animals (Fig. 2d).

Answer:

We have provided high-resolution pictures for the livers and spleens in the revised Fig. 2d.

10. In the gene expression study with DPF3 overexpression, the authors observe a high correlation with SMARCA4-controlled gene sets (Fig. 3c), but no changes in chromatin accessibility (Fig. 3d). Can they investigate accessibility changes directly at the promoters of these SMARCA4-controlled genes?

Answer:

We got the list of SMARCA4-controlled genes from Ingenuity Pathway Analysis (IPA) and then compared ATAC-seq signals at promoters of these gene between DPF3a-OE and control cells and observed no significant difference (shown below).

Figure legend: Scatter plot comparing ATAC-seq normalized read counts (CPM: counts per million) at the promoters of SMARCA4-controlled genes in 786-O cells with versus those without DPF3a overexpression. Pearson's correlation test was performed. The blue and red dash line represents fold change of 1.5 and 2 respectively.

11. The proteomics data (Fig 4a) along with previously published data show that DPF3 is excluded from PBAF complexes. Does the overexpression system simply shift the balance between BAF and PBAF complexes, thereby phenocopying a state of PBRM1 mutation?

Answer:

From our IP-MS data, it was clear that DPF3 and SNIP1 only associated with BAF components in 786-O cells (Fig. 4a and Supplementary Fig. 4b). PBRM1 was a core member of PBAF complex. It has been reported as a tumor suppressor and loss of PBRM1 in ccRCC cells promoted proliferation, migration and adhesion. As suggested by the referee, to check whether DPF3a overexpression shift the balance between BAF and PBAF, thereby phenocopying a state of PBRM1 mutation, we measured the expression level of DPF3a target genes upon PBRM1 knockdown. As shown below,

we did not observe consistent alterations, as observed in cells with DPF3a overexpression. Therefore, we believed at least the observed effect of DPF3a overexpression in this study was unlikely due to the shifting balance between BAF and PBAF complexes.

Figure legend: Expression levels of DPF3a target genes were analyzed in 786-O cells after PBRM1 knockdown.

Reviewers' Comments:

Reviewer #1:

Remarks to the Author:

The authors has put a lot of effort in trying to address the several points raised by the four reviewers - which they did using the tools they have experience with. Thus, their proposed model still lacks biophysical evidences, if not structural data, to support the "soft" data from the Co-IP and pull-down assays to demonstrate a direct protein-protein interaction between DPF3a and SNIP1, as well as the mapping of this putative interaction onto both binding partners. These are indeed not trivial studies to carry out - but they are a fundamental element to their mechanistic claims.

One minor point, the authors should be more thorough in their references - for example at page 10, Line 226 they have forgotten to cite the paper they refer to about the N-terminus not been available for making protein-protein interactions with other SWI/SNF subunits as it is buried within the chromatin remodelling complex (He et al., Science 2020).

Reviewer #2:

Remarks to the Author:

The authors have satisfactorily addressed all the comments that I raised in the last review. It is clear that the authors have made substantial efforts to address the comments from all four reviewers by furnishing significant amounts of new data. The revised manuscript has been significantly improved. Hence, I recommend the acceptance of the manuscript for publication in Nat. Common.

Reviewer #3:

Remarks to the Author:

The authors have satisfactorily addressed my previous comments.

Reviewer #4:

Remarks to the Author:

In this revised version, the authors have now addressed most of the referee comments and thereby significantly strengthened the manuscript. I now support publication in principle but still urge authors to further improve the manuscript based on the comments below:

1, The fold changes in the new Westerns Fig 3c are not particularly impressive, even less so for the MMP2/9 activity in Fig. 3d. Here in particular, it is essential to improve the method section to clearly describe how this experiment was conducted and whether it can be ruled out that this minimal change is caused by increased proliferation and therefore more secreting cells following DPF3a overexpression.

2, For all the new targeted ChIPs that mostly show less than 2-fold changes that are notoriously difficult to detect by qPCR (Fig. 6e, S7hj), a non DPF3-dependent control locus (as in 5i) should be included.

3, Similarly, for the new mass spec data in S4b, ideally also some specific DPF3a independent interactors should be shown.

4, The authors claim multiple times that there are only mild splicing alterations following SNIP1 knockdown and DPF3a overexpression . However, the number of significant splicing changes (724/2138) appear comparable to those changing gene expression (824/1713).

Point-by-point response to the referees' comments

Reviewer #1 (Remarks to the Author):

The authors has put a lot of effort in trying to address the several points raised by the four reviewers - which they did using the tools they have experience with. Thus, their proposed model still lacks biophysical evidences, if not structural data, to support the "soft" data from the Co-IP and pull-down assays to demonstrate a direct protein-protein interaction between DPF3a and SNIP1, as well as the mapping of this putative interaction onto both binding partners. These are indeed not trivial studies to carry out - but they are a fundamental element to their mechanistic claims.

One minor point, the authors should be more thorough in their references - for example at page 10, Line 226 they have forgotten to cite the paper they refer to about the N-terminus not been available for making protein-protein interactions with other SWI/SNF subunits as it is buried within the chromatin remodelling complex (He et al., Science 2020).

Answer: We appreciate the positive feedback from this referee. We have gone through the references carefully. In particular, we have added two references regarding the N-terminus of DPF3 in the SWI/SNF complexes as suggested by the referee, as well as cited the Human Cancer Metastasis Database as suggested by the editor. Please refer to reference no. 30, 31 and 50.

Reviewer #2 (Remarks to the Author):

The authors have satisfactorily addressed all the comments that I raised in the last review. It is clear that the authors have made substantial efforts to address the comments from all four reviewers by furnishing significant amounts of new data. The revised manuscript has been significantly improved. Hence, I recommend the acceptance of the manuscript for publication in Nat. Common.

Answer: We appreciate the positive feedback from this referee.

Reviewer #3 (Remarks to the Author):

The authors have satisfactorily addressed my previous comments.

Answer: We appreciate the positive feedback from this referee.

Reviewer #4 (Remarks to the Author):

In this revised version, the authors have now addressed most of the referee comments and thereby significantly strengthened the manuscript. I now support publication in principle but still urge authors to further improve the manuscript based on the comments below:

1, The fold changes in the new Westerns Fig 3c are not particularly impressive, even less so for the

MMP2/9 activity in Fig. 3d. Here in particular, it is essential to improve the method section to clearly describe how this experiment was conducted and whether it can be ruled out that this minimal change is caused by increased proliferation and therefore more secreting cells following DPF3a overexpression.

Answer: Thanks for this comment. The MMP2/9 activity was analyzed using culture medium at 24 hours after seeding, when no proliferation difference was observed. We have now described this in the method section as suggested by the referee.

2, For all the new targeted ChIPs that mostly show less than 2-fold changes that are notoriously difficult to detect by qPCR (Fig. 6e, S7hj), a non DPF3-dependent control locus (as in 5i) should be included.

Answer: As suggested by the referee, we have now added ACTB, a non DPF3-dependent locus in Fig. 6e, Supplementary Fig. 7h&i.

3, Similarly, for the new mass spec data in S4b, ideally also some specific DPF3a independent interactors should be shown.

Answer: As suggested by the referee, we have now included SNIP1 interaction partners that were independent of DPF3a in Supplementary Table 1.

4, The authors claim multiple times that there are only mild splicing alterations following SNIP1 knockdown and DPF3a overexpression. However, the number of significant splicing changes (724/2138) appear comparable to those changing gene expression (824/1713).

Answer: It was inappropriate to simply compare the numbers of splicing changes with gene expression alterations, as the total numbers of splicing events were much more compared to that of expressed genes. Upon SNIP1 knockdown and DPF3a overexpression, 724 out of 90263 (0.8%) and 2138 out of 82073 (2.6%) spliced events were differentially spliced, respectively. In contrast, 824 out of 12037 (6.8%) and 1713 out of 11610 (14.8%) genes were differentially expressed after SNIP1 knockdown and DPF3a overexpression, respectively.